# THE ILLUSION OF FORGETTING: POST-HOC UTILITY RECOVERY FROM UNLEARNED MODELS

## ABSTRACT

Class unlearning seeks to remove the influence of designated training classes while retaining utility on the remaining ones, often for privacy or regulatory compliance. Existing evaluations largely declare success once the forgotten classes exhibit near-zero accuracy or fail membership inference tests. We argue this view is incomplete and introduce the notion of *the illusion of forgetting*: even when accuracy appears suppressed, the black-box outputs of unlearned models can retain residual, recoverable signals about forgotten classes. We formalize this phenomenon by quantifying residual information in the output space and show that unlearning trajectories leave statistically distinguishable signatures. To demonstrate practical implications, we propose a simple yet effective post-hoc recovery framework, which amplifies weak signals using a Yeo–Johnson transformation and adapts decision thresholds to reconstruct predictions for forgotten classes. Across 12 unlearning algorithms and 4 benchmark datasets, our framework substantially restores forgotten-class accuracy while causing minimal degradation on retained classes. These findings (i) expose critical blind spots in current unlearning evaluations, (ii) provide the first systematic evidence that forgotten-class utility can be restored from black-box access alone.

## 1 INTRODUCTION

Removing the influence of specific training classes from a deployed model is increasingly required for privacy compliance, regulatory mandates, and user expectations. Recent class unlearning methods demonstrate impressive performance, often driving forgotten classes' accuracy towards zero while maintaining utility on the remaining data Kurmanji et al. (2023); Shokri et al. (2017); Graves et al. (2021); Chen et al. (2024; 2021).

This apparent success has fueled the belief that low forgotten-class accuracy or failed membership inference is sufficient to guarantee forgetting.

We show that this view is incomplete. Even after passing such standard forgetting checks, unlearned models can still retain residual, recoverable signals about the forgotten classes. **Why this matters.** Consider a medical classifier where a hospital requests the removal of a rare disease class to comply with privacy regulations. Conventional metrics may certify "success" once forgotten-class accuracy is suppressed to chance. Yet if a downstream observer can still recover diagnostic capability for the removed disease, the deletion request—and the hospital's privacy rights—would be undermined. In this work, we make a central observation: unlearned models often leave weak but structured traces of the

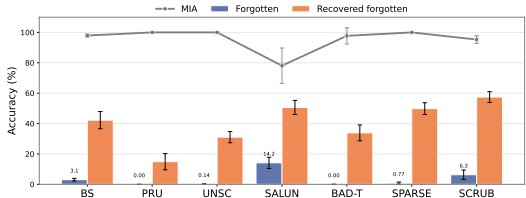

Figure 1: The *illusion of forgetting*. Standard evaluations report forgotten-class accuracy near zero after unlearning, suggesting success. However, a lightweight recovery procedure can restore substantial utility to the forgotten classes. The figure reports average results over seven representative unlearning methods Chen et al. (2023); Zhang et al. (2024); Chen et al. (2024); Fan et al. (2024); Chundawat et al. (2023a); Jia et al. (2023); Kurmanji et al. (2023).

forgotten classes in their output probabilities. With lightweight post-processing, these traces can be amplified into meaningful re-predictions. We call this gap between apparent forgetting and residual recoverable utility the *illusion of forgetting*. See Figure 1 for an illustration, where forgotten-class accuracy falls to near zero under unlearning but can be substantially restored.

Why does recovery remain possible after "successful" unlearning? Our theoretical analysis formalizes two intuitions. First, *retained-class utility and complete forgetting are in tension*: preserving high accuracy on the remaining classes generally necessitates retaining structured information about forgotten classes in the representation and output space. Second, unlearning trajectories leave *statistical signatures in output space*: to suppress forgotten classes, algorithms collapse their outputs to narrow ranges near zero, which still encode distinguishable patterns.

To demonstrate the practical severity of this phenomenon, we introduce a simple black-box recovery framework. It assumes only realistic access to the deployed unlearned model's softmax probabilities, without weights, gradients, or training data. The framework applies a monotone statistical transform (Yeo–Johnson Weisberg (2001)) to stabilize and amplify weak near-zero outputs, followed by adaptive thresholding to map transformed scores back into forgotten-class predictions. This procedure is model-agnostic, requires no retraining, and serves to reveal recoverability rather than to propose yet another unlearning algorithm. Comprehensive experiments across 12 state-of-the-art unlearning methods and 4 benchmark datasets show that forgotten-class utility can be recovered far beyond random guessing while minimally affecting retained-class performance. These findings expose a critical blind spot in current unlearning evaluation and highlight the urgent need to reconsider what forgetting truly guarantees.

Our contributions are:

- We uncover and formalize the *illusion of forgetting*, i.e., the gap between certified forgetting and residual, recoverable utility, revealing a blind spot in how class unlearning is currently evaluated.

- Through theoretical analysis, we show that preserving retained-class accuracy inherently leaves structured traces of forgotten classes, and that typical unlearning trajectories produce distinctive statistical signatures in the output space.

- To make the phenomenon tangible, we design a lightweight black-box recovery framework (based on Yeo–Johnson transformation and adaptive thresholding) that converts weak residual signals into forgotten-class predictions—without accessing weights, gradients, training data, or any labels.

- Extensive experiments across 12 state-of-the-art unlearning methods and 4 benchmark datasets reveal that most are susceptible to the illusion of forgetting, enabling recovery well beyond random guessing while retaining performance on the remaining classes.

## 2 BACKGROUND

Let $\mathcal{X} \subset \mathbb{R}^d$ denote the input space and $\mathcal{Y} = \{1, \ldots, K\}$ denote the label space with $K$ classes. The training dataset $\mathcal{D} = \{(x_i, y_i)\}_{i=1}^N$ contains $N$ samples.

In class machine unlearning, the objective is to eliminate the influence of all training samples from a target class $c \in \mathcal{C} \subset \mathcal{Y}$. This naturally partitions the dataset into the forgetting set $\mathcal{D}_f = \{(x_i, y_i) \in \mathcal{D} : y_i \in \mathcal{C}\}$ containing samples to be unlearned and the retaining set $\mathcal{D}_r = \mathcal{D} \setminus \mathcal{D}_f$ containing samples to be preserved.

**Definition 1** (*Unlearning Process*) *Let $f_{\theta_0} : \mathcal{X} \to \mathbb{R}^K$ be the original model with parameters $\theta_0$. The unlearning process is defined as the optimization Bourtoule et al. (2021); Graves et al. (2021):*

$$\theta^* = \arg \min_\theta \mathcal{L}_{unlearn}(\theta_0) = \mathcal{L}_r(\theta; \mathcal{D}_r) + \lambda \mathcal{L}_f(\theta; \mathcal{D}_f) \tag{1}$$

*where $\mathcal{L}_r$ preserves performance on retained data and $\mathcal{L}_f$ enforces forgetting on $\mathcal{D}_f$.*

### 2.1 EVALUATION PARADIGMS

The existing works in MU primarily evaluate forgetting effectiveness through various accuracy-based metrics Chen et al. (2024); Graves et al. (2021), such as unlearning accuracy ($UA$)

and retaining accuracy ($RA$):

$$UA = \frac{1}{\|\mathcal{D}_f\|} \sum_{(x,y) \in \mathcal{D}_f} \mathbb{I}[f_{\theta^*}(x) = y]; \quad RA = \frac{1}{\|\mathcal{D}_r\|} \sum_{(x,y) \in \mathcal{D}_r} \mathbb{I}[f_{\theta^*}(x) = y].$$

Compare $UA$ and $RA$ on the unlearned model with the results on $f_{\theta^*}$ to assess whether the unlearning algorithms are successful or not. Commonly, success is declared when $UA \approx 0$ and $RA \approx RA(f_{\theta^0}, \mathcal{D}_r)$ Graves et al. (2021); Bourtoule et al. (2021); Chen et al. (2023); Foster et al. (2024).

In addition to accuracy-based metrics, some recent works employ MIA Shokri et al. (2017); Chen et al. (2021) to verify that forgotten samples cannot be distinguished from unseen data, while others test forgetting quality by measuring how quickly the unlearned model can relearn the forgotten class through fine-tuning on $D_f$ Chundawat et al. (2023b); Golatkar et al. (2020).

However, all these current evaluation paradigms share a common focus on: seeking to answer the question Are the unlearning samples truly forgotten?. There still is a gap to *Can the utility on forgotten classes be recovered through only **an unlearned model**?*.

## 2.2 Attack on Machine Unlearning

Recent research has identified various attack vectors against unlearning systems. Adversarial approaches exploit the unlearning mechanism to trigger hidden behaviors Di et al. (2022); Liu et al. (2024) or lag the unlearning to increase computational costs Marchant et al. (2022). Privacy-focused attacks leverage information leakage to infer training information Chen et al. (2021); Gao et al. (2022); Lu et al. (2022); Hu et al. (2024). Unlike prior attacks typically require access to original models Hu et al. (2024); Lu et al. (2022); Chen et al. (2021); Gao et al. (2022) or involve model modifications Xiao et al. (2025), our approach differs fundamentally in both methodology and objective. Rather than attempting adversarial exploitation, we demonstrate that utility recovery on forgotten classes is possible through statistical analysis of black-box outputs alone, revealing inherent limitations in class unlearning. This suggests that current unlearning processes may not fully achieve their fundamental objective of removing sample influence, a gap we systematically investigate in this work.

## 2.3 Recovery Evaluation

We define how the recovery is evaluated in this work. Given a test dataset $\mathcal{D}_{test}$, the utility of the unlearned model $f_{\theta^*}$ for the forgotten class $c$ can be evaluated by the following metrics:

$$UA(g \circ f_{\theta^*}, \mathcal{D}_f^{test}) > 1/K \quad \text{and} \quad RA^* - \Delta \leq RA(g \circ f_{\theta^*}, \mathcal{D}_r^{test}) \leq RA^* + \Delta, \quad (2)$$

where $\Delta$ denotes an acceptable change in the accuracy.

The unlearned model that can be deployed to users can obtain positive gain via the processing method $g$, which indicates the successful utility recovery. If utility recovery succeeds under these minimal assumptions, it reveals that the unlearned model retains sufficient residual information about the forgotten class to enable classification, contradicting the privacy guarantees that unlearning aims to provide.

## 3 Method

In this section, firstly, we present the theoretical analysis to explain why machine unlearning inevitably leaves recoverable traces, then propose a post-hoc recovery framework that exposes these vulnerabilities through statistical analysis of output probability distributions.

### 3.1 Theoretical Analysis

To understand why machine unlearning algorithms inevitably leave recoverable traces of forgotten information, we analyze the fundamental trade-offs between forgetting completeness and utility preservation. We establish information-theoretic bounds on residual information (Theorem 1),

characterize the geometric properties of class-specific forgetting trajectories (Theorem 2), and quantify the statistical detectability of these residual patterns (Theorem 3). The detailed proofs of all theorems are presented in Appendix A.

**Definition 2** *(Residual Information) For an unlearned model $f_{\theta^*}$ and forgotten class c, the residual information is:*

$$\mathcal{I}_{res}(c) = I(f_{\theta^*}(X); Y_c \mid X \in \mathcal{X}_c), \tag{3}$$

*where $I(\cdot; \cdot)$ denotes mutual information Cover (1999), $\mathcal{X}_c$ is the input distribution of class c, and $Y_c$ is the label indicator for the forgotten class c.*

### 3.1.1 INFORMATION-THEORETIC ANALYSIS

We first delve into complete information removal, which could be impossible when maintaining utility on retained classes. This reveals a conflict between forgetting specific information and preserving model capabilities.

**Theorem 1** *(Incompatibility of Complete Forgetting and Utility Preservation) For any unlearning algorithm $\mathcal{A}$ that maintains accuracy $\alpha_r \geq \alpha_0$ on remaining classes, the residual information satisfies:*

$$\mathcal{I}_{res}(c) \geq \frac{1}{K}\Big[ \log K - H\left(\frac{1 - \alpha_0}{K - 1}\right) - (1 - \alpha_0)\log(K - 1)\Big], \tag{4}$$

*where $\alpha_r$ and $\alpha_0$ are the accuracy of the unlearned model $f_{\theta^*}$ and the original model $f_{\theta_0}$ on remaining classes respectively, and $H(\cdot)$ is the entropy function.*

This theorem reveals a fundamental trade-off: maintaining high accuracy on retained classes generally leaves residual information about forgotten classes. The bound increases with a higher original accuracy $\alpha_0$, as preserving performance requires shared representations that cannot be completely disentangled. This explains why existing unlearning methods showing good forgetting performance may still contain recoverable information.

### 3.1.2 GEOMETRIC ANALYSIS OF FORGETTING TRAJECTORIES

To better understand the source of residual information, we now investigate how this information manifests in the parameter space.

**Theorem 2** *(Class-Specific Forgetting Trajectories) For distinct classes $c_i, c_j \in \mathcal{Y}$, their forgetting trajectories $\gamma_{c_i}(t), \gamma_{c_j}(t)$ satisfy:*

$$\mathbb{E}[\|\gamma_{c_i}(t) - \gamma_{c_j}(t)\|_2] \geq \delta(t) \cdot \sqrt{\frac{d_{c_i} + d_{c_j}}{2K}}, \tag{5}$$

*where $\delta(t)$ is monotonically increasing for small t with $\delta(0) = 0$ and $d_{c_i}, d_{c_j}$ are the dimensionalities of the features.*

This separation of trajectories occurs because each class has a unique data distribution that creates distinct gradient patterns during unlearning. The divergence between paths depends on the distributional differences between classes, where more distinct class distributions lead to greater trajectory separation. This provides the theoretical foundation for identifying which specific class was forgotten by analyzing the unlearned model's behavior.

### 3.1.3 STATISTICAL DETECTABILITY OF RESIDUAL PATTERNS

The existence of residual information and distinguishable patterns raises a critical question about practical detectability. We now quantify the sample complexity required to reliably recover the forgotten class information from finite observations.

**Theorem 3** *(Recovery Success Bound) Let $P[recovery]$ denote the probability that the recovery accuracy on forgotten class c exceeds random guessing by margin $\varepsilon$, i.e., $P[Accuracy > 1/K + \varepsilon]$. Given n test samples and residual information $\mathcal{I}_{res}(c) > 0$, the probability of successful recovery satisfies:*

$$P[recovery] \geq 1 - \exp(-2n \cdot \mathcal{I}_{res}(c)) \tag{6}$$

This exponential relationship between sample size and recovery probability has important practical implications. Even with small residual information, recovery becomes highly probable with sufficient samples. This explains why we can successfully recover across diverse unlearning algorithms. Although the traceable signal is weak, it is consistently detectable with reasonable sample sizes.

Our theoretical framework reveals why the "illusion of forgetting" is inevitable in machine unlearning. The vulnerability we identify is not a flaw in specific unlearning algorithms but an inherent consequence of how neural networks encode and share information across classes.

## 3.2 POST-HOC UTILITY RECOVERY FRAMEWORK

Based on our theoretical insights, we propose the Post-hoc Utility Recovery framework method to exploit residual information in unlearned models. Specifically, the framework operates through statistical analysis of output probability distributions in four steps: 1) forgotten class identification, 2) probability distribution extraction & re-scaling, 3) adaptive threshold determination, and 4) threshold-based Re-prediction.

### 3.2.1 FORGOTTEN CLASS IDENTIFICATION

In real-world practical scenarios, the downstream model users could not know the unlearned classes. Therefore, we identify potential forgotten classes through forgotten class detection in the output distribution. For each output neuron node $k \in \{1, ..., K\}$, we compute the average probability and variance on the test dataset, for $x \in \mathcal{D}_{test}$:

$$\bar{p}_k = \frac{1}{|\mathcal{D}_{test}|} \sum_{x \in \mathcal{D}_{test}} p_k(x), \tag{7}$$

where $p_k(x) = S(f_{\theta^*}(x))_k$ and $S$ denotes softmax function. For an initialized model on balanced data, it expects $\bar{p}_k \approx 1/K$ for all classes. However, forgotten classes exhibit a distinctive characteristic: near-zero average probability, typically $\bar{p}_c \ll 1/K$, as the unlearning process squeezes all outputs for the forgotten class to a range near zero. Based on the forgotten class detection strategy, we identify the set of potentially forgotten classes as:

$$\mathcal{C}_f = \{k : \bar{p}_k < \kappa/K\}, \tag{8}$$

where $\kappa \in (0, 1)$ is a fixed small scaling constant.

### 3.2.2 PROBABILITY DISTRIBUTION EXTRACTION & RE-SCALING

After identifying forgotten classes $\mathcal{C}_f$, we extract their output probabilities for the downstream decision. Post-unlearning, the forgotten-class probabilities typically *concentrate near zero* with heavy skew and heteroscedastic tails. Direct thresholding on raw $p$ can therefore be unstable: tiny numeric fluctuations around 0 dominate the decision boundary and become sensitive to skew and scale. To this end, it is necessary to seek a data transformation that should (i) preserve ranking, which thus does not change the fixed-quantile operating point, while (ii) correcting skew and stabilizing within-class variance so that unsupervised thresholding is more reliable at finite sample sizes. Therefore, we apply the Yeo-Johnson transformation Weisberg (2001), which can preserve the order but pulls apart the near-zero region and reduces tail sensitivity, which will benefit unsupervised threshold estimation stability. Formally, the probability value of $p \in \{p_c(x_1), p_c(x_2), ..., p_c(x_{N_t})\}$, where $N_t$ denotes the number of test examples, the transformation mapping can be formalized as:

$$\mathcal{T}_\lambda(p) = \begin{cases} \frac{(p+1)^\lambda - 1}{\lambda} & \text{if } \lambda \neq 0, \\ \log(p+1) & \text{if } \lambda = 0, \end{cases} \tag{9}$$

where $\lambda$ is the transformation parameter, and the optimal parameter $\lambda^*$ is selected by maximizing the Gaussian log-likelihood estimation without any labels:

$$\lambda^* = \arg\max_\lambda \sum_{i=1}^n \left[ \log \phi \left( \frac{\mathcal{T}_\lambda(p_i) - \bar{\mu}_\lambda}{\sigma_\lambda} \right) - \log \sigma_\lambda + \log \left| \frac{d\mathcal{T}_\lambda}{dp} \Big|_{p=p_i} \right| \right], \tag{10}$$

where $\phi$ is the standard normal PDF, and $\bar{\mu}_\lambda$, $\sigma_\lambda$ are the sample mean and standard deviation of transformed values.

### 3.2.3 ADAPTIVE THRESHOLD DETERMINATION

The key insight is that different forgotten classes exhibit distinct patterns in this transformed space due to their class-specific unlearning trace, i.e., Theorem 2.

Given a set of transformed probability values $\{\mathcal{T}_\lambda(p_i)\}$ for the output corresponding to forgotten class, regarded as binary classification problems (e.g., $x \in \mathcal{D}'_{test}$ or $x \notin \mathcal{D}'_{test}$), we apply Otsu's method Otsu (1975) adaptive threshold determination method to find the optimal threshold $\tau^*$ that maximizes the between-class variance:

$$\tau^* = \arg\max_\tau \sigma_B^2(\tau), \tag{11}$$

where the between-class variance $\sigma_B^2(\tau)$ is:

$$\sigma_B^2(\tau) = \omega_0(\tau)\omega_1(\tau)[\mu_0(\tau) - \mu_1(\tau)]^2, \tag{12}$$

where $\omega_0(\tau)$ and $\omega_1(\tau)$ are the proportions of samples below and above threshold $\tau$, with corresponding means $\mu_0(\tau)$ and $\mu_1(\tau)$. For the adaptive thresholding, Gaussian Mixture Models (GMM) Reynolds (2015) and k-means clustering can also be utilised as an alternative method.

### 3.2.4 RE-PREDICTION

Once we obtain the transformed probability $\mathcal{T}_{\lambda^*}(p(x))$ and the optimal threshold, we can execute the final classification for $\mathrm{D}_{test}$, which is defined as:

$$\hat{y}(x) = \begin{cases} c & \text{if } \mathcal{T}_{\lambda^*}(p_c(x)) > \tau^*, \\ \arg\max_{j \neq c} S(f_{\theta^*}(x))_j & \text{otherwise.} \end{cases} \tag{13}$$

The classification rule allows us to recover the model's performance from the unlearned model. The summary of the framework is shown in Algorithm 1.

### 3.2.5 EXTENSION TO MULTI-CLASS FORGETTING

The framework can be easily extended to a multi-class forgetting scenario. After the identification of the forgotten classes, we aggregate probabilities across $\mathcal{C}_f$, instead of analyzing each $c \in \mathcal{C}_f$ separately. For each sample $x \in \mathcal{D}_{test}$, aggregate the probabilities across $\mathcal{C}_f$:

$$p_{agg}(x) = \sum_{c \in \mathcal{C}_f} p_c(x). \tag{14}$$

This aggregation captures the total response to potentially forgotten samples. Apply the transformation and threshold determination following in Step 2 3.2.2 and Step 3 3.2.3 to $p_{agg}(x)$ with optimal parameter $\lambda^*_{agg}$ and $\tau^*_{agg}$.

Finally, we can re-predict on $\mathcal{D}_{test}$ following the classification rules:

$$\hat{y}(x) = \begin{cases} \arg\max_{c \in \mathcal{C}_f} p_c(x) & \text{if } \mathcal{T}_{\lambda^*}(p_{agg}(x)) > \tau^*_{agg} \\ \arg\max_{j \notin \mathcal{C}_f} S(f_{\theta^*}(x))_j & \text{otherwise.} \end{cases} \tag{15}$$

## 4 EXPERIMENTS

In this section, we empirically demonstrate the effectiveness of our post-hoc utility recovery framework across diverse unlearning methods and datasets.

### 4.1 EXPERIMENTAL SETUP

**Datasets & backbone models.** We conduct experiments on four benchmark datasets with varying complexity: *MNIST* LeCun et al. (1998), *FMNIST* Xiao et al. (2017), *CIFAR-10* Krizhevsky &

Hinton (2009), and *CIFAR-100* Krizhevsky & Hinton (2009). For each dataset, we randomly sample 80% as the training set and 20% as the validation set from the original datasets. The test dataset is used from the officially provided test set.

Following the existing works Chen et al. (2024); Chundawat et al. (2023b), we adopt *AllCNN* Springenberg et al. (2014) as the backbone for MNIST and FMNIST, *ResNet18* He et al. (2016) for CIFAR10 and *ResNet34* He et al. (2016) for CIFAR-100. All models are trained to converge on the original dataset before unlearning, details shown in Appendix D.1.

**Unlearning Methods.** We evaluate 12 state-of-the-art unlearning methods spanning different approaches. The approaches includes *Unroll* Thudi et al. (2022), *Unroll-F* Thudi et al. (2022), *GA* Golatkar et al. (2020), *Fisher* Golatkar et al. (2020), Boundary Shrink (*BS*) Chen et al. (2023), Boundary Expand (*BE*) Chen et al. (2023), *Bad-T* Chundawat et al. (2023a), *SPARSE (SP)* Jia et al. (2023), *SCRUB* Kurmanji et al. (2023), *SALUN* Fan et al. (2024), *UNSC* Chen et al. (2024) and *PRU* Zhang et al. (2024). RT is a fundamental method of retraining from scratch on the remaining classes' data. Since the RT model is trained only on the remaining classes, the model has corresponding classification nodes, without the nodes corresponding to the forgotten classes. The RT original model comes with initialized classifier nodes for evaluation on the forgetting data $\mathcal{D}_f$. More details are shown in Appendix D.2.

**Unlearning & Recovery Tasks** We evaluate our post-hoc recovery framework across single-class and multi-class unlearning scenarios. For each method-dataset combination, we first train the original model $f_{\theta_0}$ on the complete dataset $\mathcal{D}$, then apply the unlearning algorithm to forget target classes, producing unlearned models $f_{\theta^*}$. For single-class scenarios, we systematically forget each class individually, providing $K$ experimental target classes per method-dataset combination. For multi-class scenarios, we randomly select two target classes for MNIST, FMNIST, and CIFAR-10, and 10 classes for CIFAR-100. Each experiment is repeated three times with different random seeds, i.e.,$\{0, 1, 2\}$, to ensure statistical reliability.

**Evaluation Metrics.** We report accuracies on forgotten classes (FA) and on remaining classes (RA) using the *unlearned* model and the FA and RA results after post-hoc restoration processing. This directly reflects the two desiderata of class unlearning: (i) removing utility on the forgotten classes and (ii) preserving utility on the remaining classes. In Tab. 15 and 1, the upper line of each method shows the unlearned results; the lower line with a gray background shows the post-hoc restoration utility results. The MIA results to show the unlearning efficacy are shown in Tab. 18.

## 4.2 MAIN RESULTS

**Utility restoration under single- and multi-class unlearning.** Tabs. 1 and 15 report results on four benchmarks and a broad set of unlearning methods. As expected, unlearning suppresses forgotten-class accuracy (FA) while largely preserving the accuracy on remaining classes (RA), and the MIA score (reported as TNR) indicates successful erasure under standard criteria. After applying our post-hoc recovery, however, FA rises by tens of percentage points, whereas RA changes only marginally (typically $< 5\%$), and this trend holds on all datasets and across unlearning approaches. In several cases, we even restore strong forgotten-class performance when the corresponding unlearned FA is exactly zero, showing that "near-zero accuracy" alone can be misleading.

In the single-class scenario 15, when unlearning preserves high RA, the model must retain shared representations that encode information about the forgotten class; our theory formalizes this trade-off and lower-bounds the residual information $I_{res}$ in such scenarios. Consequently, the probability of successful restoration increases exponentially with both the available samples and $I_{res}$, matching the consistent FA gains we observe. In the multi-class scenario, Tab. 1 shows the same qualitative pattern: large FA gains after recovery with small RA changes. The level of restored FA depends on the number of forgotten classes and their semantic proximity, but the conclusion remains—the forgotten subset's utility is systematically recoverable from the output space of $f_{\theta^*}$.

We identify two main failure modes consistent with our analysis and the method designs. (i) *Representation collapse.* If unlearning also degrades RA, the retained features become less informative, shrinking $I_{res}$ and reducing detectability; this follows directly from the information-utility trade-off. (ii) *Structural constraints at the head.* RT retrains from scratch on

Table 1: **Unlearning and Recovery on multi-class forgetting** (mean $\pm$ std over three runs). Upper line: original unlearned results; lower line (with gray background): Post-hoc recovery results. FA denotes accuracy on forgotten classes; RA denotes accuracy on remaining classes.

| Method | MNIST | | FMNIST | | CIFAR-10 | | CIFAR-100 | |
|---|---|---|---|---|---|---|---|---|
| | FA | RA | FA | RA | FA | RA | FA | RA |
| Orig | 99.52±0.15 | 99.56±0.04 | 91.94±4.87 | 93.27±1.22 | 91.82±2.52 | 93.25±0.63 | 73.04±3.93 | 72.01±0.44 |
| RT | 0.00±0.00 | 99.63±0.10 | 0.00±0.00 | 94.31±1.77 | 0.00±0.00 | 94.07±0.85 | 0.00±0.00 | 72.78±0.67 |
| | 47.27±11.73 | 98.29±0.59 | 25.62±10.66 | 89.96±1.66 | 18.89±5.30 | 89.72±0.90 | 2.88±0.56 | 69.15±0.53 |
| PRU | 0.00±0.00 | 99.43±0.21 | 0.08±0.09 | 94.24±1.64 | 0.00±0.00 | 94.33±0.91 | 0.00±0.02 | 71.14±0.66 |
| | 49.37±10.15 | 99.40±0.21 | 44.71±7.46 | 93.46±1.39 | 56.59±7.28 | 90.95±0.77 | 14.90±5.33 | 66.01±0.72 |
| BS | 0.69±1.90 | 74.75±13.06 | 1.19±2.26 | 90.51±3.17 | 0.92±0.67 | 89.13±0.91 | 3.07±0.87 | 61.15±1.58 |
| | 46.99±21.55 | 69.18±13.66 | 14.66±13.25 | 89.24±3.55 | 16.08±5.48 | 84.68±1.55 | 55.79±1.28 | 57.79±1.28 |
| BE | 8.77±5.19 | 94.84±2.43 | 19.15±7.15 | 91.65±3.25 | 39.68±3.30 | 93.27±0.89 | 35.84±3.45 | 70.38±0.60 |
| | 47.57±8.29 | 94.20±2.83 | 58.78±7.49 | 91.28±3.33 | 83.92±3.30 | 92.67±0.71 | 77.22±3.25 | 67.08±0.32 |
| Unroll | 0.00±0.00 | 55.70±20.97 | 2.35±8.32 | 70.90±22.86 | 49.27±8.35 | 93.12±1.28 | 59.58±4.83 | 72.05±0.55 |
| | 13.22±9.66 | 52.42±19.38 | 50.08±19.09 | 62.29±22.91 | 67.58±6.25 | 92.81±1.21 | 73.27±3.72 | 70.90±0.46 |
| Unroll-F | 20.16±19.55 | 97.35±2.30 | 3.31±4.93 | 87.93±3.90 | 29.38±11.93 | 90.15±2.11 | 40.93±6.78 | 70.08±1.10 |
| | 95.00±2.98 | 97.31±2.29 | 36.00±9.34 | 87.73±3.94 | 78.18±8.42 | 88.82±1.46 | 77.44±5.24 | 66.25±0.61 |
| UNSC | 0.00±0.02 | 99.55±0.09 | 0.15±0.11 | 94.17±1.49 | 0.00±0.00 | 94.30±0.86 | 0.14±0.14 | 72.88±0.56 |
| | 64.51±3.77 | 98.55±0.59 | 48.85±10.53 | 90.73±2.09 | 28.64±3.39 | 93.17±0.92 | 31.08±3.71 | 72.41±0.59 |
| SALUN | 0.03±0.04 | 98.00±1.65 | 0.07±0.11 | 92.63±1.35 | 0.09±0.16 | 82.43±5.00 | 14.15±3.68 | 43.59±5.73 |
| | 95.49±4.23 | 97.78±1.67 | 84.42±10.35 | 87.71±2.66 | 58.74±10.94 | 77.56±7.74 | 50.61±4.64 | 40.58±7.12 |
| GA | 20.15±13.39 | 93.63±6.55 | 6.27±9.39 | 87.53±10.02 | 0.24±0.28 | 81.25±3.87 | 8.14±10.19 | 52.34±12.70 |
| | 64.00±32.87 | 92.98±6.94 | 10.07±12.46 | 86.32±10.96 | 1.06±0.75 | 80.92±3.93 | 32.02±17.48 | 48.24±13.05 |
| Fisher | 3.50±6.86 | 98.94±0.69 | 3.42±5.48 | 93.22±1.98 | 3.64±3.46 | 93.78±0.85 | 0.00±0.00 | 71.22±0.58 |
| | 52.15±27.98 | 97.49±1.57 | 38.80±22.76 | 91.74±1.70 | 41.53±16.55 | 92.59±0.76 | 4.41±1.52 | 67.12±0.47 |
| Bad-T | 0.02±0.09 | 98.06±1.10 | 2.44±4.47 | 91.43±2.38 | 1.45±0.89 | 91.77±1.40 | 0.00±0.00 | 58.99±1.80 |
| | 51.47±8.12 | 93.93±1.99 | 46.84±10.75 | 85.98±2.31 | 53.71±5.90 | 89.75±1.42 | 33.90±5.19 | 57.17±1.81 |
| SP | 0.00±0.00 | 99.52±0.07 | 0.00±0.00 | 93.67±1.90 | 0.00±0.00 | 93.63±0.95 | 0.77±0.72 | 71.58±0.66 |
| | 85.54±5.51 | 98.62±0.31 | 53.13±10.99 | 87.75±2.49 | 51.31±6.45 | 88.30±1.27 | 49.83±3.88 | 70.26±0.55 |
| SCRUB | 2.27±4.31 | 99.02±0.28 | 2.01±2.92 | 92.54±2.19 | 0.01±0.03 | 91.12±1.27 | 6.32±3.12 | 67.68±0.69 |
| | 92.96±4.19 | 98.81±0.36 | 67.76±13.99 | 91.24±3.33 | 79.80±2.51 | 87.30±1.70 | 57.46±3.59 | 66.34±0.66 |

remaining classes and omits classifier nodes for forgotten classes in the deployed model, which forces near-zero logits for $C_f$ and limits any black-box restoration. Fisher Golatkar et al. (2020) explicitly modifies the last-layer head for forgotten classes, further clamping outputs near zero and diminishing separability after transformation.

Table 2: Comparing the Recall among our method and OOD detection methods on **CIFAR-100**.

| Method | PRU | BS | BE | Unroll | Unroll-F | UNSC | SALUN | GA | Bad-T | SP | SCRUB |
|---|---|---|---|---|---|---|---|---|---|---|---|
| MSP | 3.92±1.39 | 7.86±1.59 | 15.44±0.39 | 11.65±0.70 | 13.46±0.67 | 15.30±1.18 | 2.39±1.06 | 11.16±0.62 | 10.49±1.05 | 13.74±0.86 | 12.40±0.65 |
| Ratio | 4.11±1.41 | 7.76±1.50 | 14.61±0.49 | 11.48±0.84 | 12.98±0.85 | 14.79±1.12 | 2.44±1.16 | 10.74±0.75 | 9.73±1.17 | 14.40±1.01 | 13.52±0.81 |
| Energy | 3.16±1.10 | 6.45±1.25 | 12.68±0.54 | 10.94±0.80 | 11.87±0.73 | 12.13±0.79 | 2.24±1.14 | 11.12±0.76 | 8.32±0.78 | 11.33±0.67 | 10.74±0.69 |
| Margin | 3.77±1.33 | 7.50±1.51 | 14.83±0.44 | 11.50±0.69 | 13.05±0.63 | 14.89±1.11 | 2.37±1.04 | 10.96±0.65 | 9.68±0.97 | 13.25±0.83 | 11.99±0.61 |
| Gini | 3.57±1.28 | 7.34±1.46 | 14.82±0.42 | 11.53±0.63 | 13.05±0.57 | 14.86±1.00 | 2.35±0.99 | 11.07±0.59 | 9.50±0.93 | 12.66±0.72 | 11.40±0.53 |
| Energy+ | 3.38±1.22 | 7.29±1.40 | 15.27±0.36 | 11.70±0.53 | 13.47±0.48 | 15.29±1.13 | 2.36±1.03 | 11.80±0.49 | 10.29±1.00 | 12.38±0.63 | 11.30±0.58 |
| Entropy | 4.04±1.43 | 8.45±1.72 | 16.81±0.40 | 11.89±0.78 | 14.30±0.74 | 16.40±1.32 | 2.37±1.05 | 11.73±0.69 | 12.98±1.24 | 14.26±0.95 | 12.55±0.74 |
| **Ours** | **7.80±3.31** | **14.46±5.31** | **94.74±1.89** | **64.23±4.76** | **71.04±1.47** | **66.03±7.13** | **34.42±1.41** | **19.61±1.15** | **29.60±6.54** | **58.55±1.14** | **62.38±1.06** |

**Comparison to other possible restoration methods.** We introduce six OOD detection methods: Max Softmax Probability (MSP) Hendrycks & Gimpel (2017), Entropy-based detection Malinin & Gales (2018), Gini coefficient Liu et al. (2023), likelihood ratio Ren et al. (2019), and Energy Score-based methods (including overall-based (Energy+) Liu et al. (2020) and specific-based (Energy) Wang et al. (2021)). The details of these methods are provided in Appendix D.3. For each method, we report the Recall rate for the forgotten class, which is the ratio of correctly re-predicted samples to the total number of forgotten class samples on the test set. The results evaluated on CIFAR-100 are shown in Tab. 2, and the complete results in 16. We observe that our method achieves the best performance among all compared methods.

**How does the probability distribution re-scaling work?** As illustrated in Fig. 2, we analyze the probability distributions for the unlearned class (class 1) on the CIFAR-10 test set, comparing the original model, the model after unlearning with the SCRUB approach, and the re-scaled probability distribution. We observe that the unlearning causes the probability distribution of the forgotten class to be compressed into a narrow range near zero in Fig. 2b. This is the reason why the

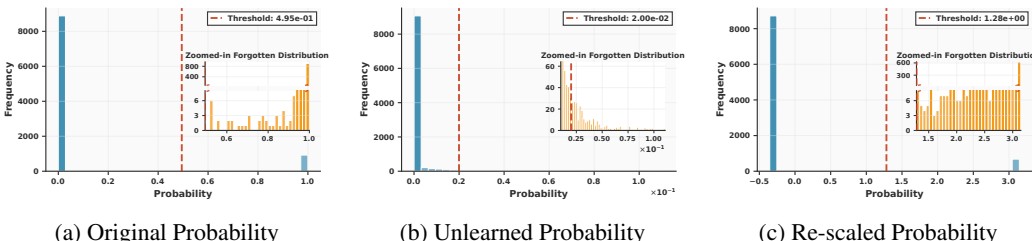

(a) Original Probability      (b) Unlearned Probability      (c) Re-scaled Probability

Figure 2: All samples' probability distribution for the unlearned class (class 1) on the test-set of CIFAR-10. We compare the original model, the model after unlearning with **SCRUB**, and its re-scaled probability distribution.

unlearning signifies an apparent success in unlearning, where the model loses its classification ability, and consequently, MIAs consistently classify these samples as non-members, as their outputs are indistinguishable from true unseen data. This state represents a fragile form of unlearning, as well as the illusion of forgetting. This illusive forgetting is not a result of true knowledge removal, but an artifact of the compressed output space, which still leaves significant traceable clues. As shown in Fig. 2c, the transformation process rescales the distribution to a narrower range, re-enabling the fundamental clustering assumption and enabling the previously forgotten class samples to be distinguished once again.

**How do data conditions impact utility recovery?** We assess the robustness of unlearning methods against class imbalance and reduced test-set size. The assessment is conducted on CIFAR-100 with ten forgetting classes. We control class imbalance via a ratio $\rho$, capping each unlearn class at $\lfloor \rho \bar{n} \rfloor$ samples, where $\bar{n} = \lfloor N/C \rfloor$ is the average per-class size. Figure 3a reveals a key trade-off: low imbalance ratios result in poor forgetting efficacy and, paradoxically, degraded retain-set accuracy. We attribute this to the model misclassifying scarce samples of forgotten classes into the more dominant retain classes. As $\rho$ increases, forgetting efficacy improves and the accuracy on remaining classes recovers, with a maximal degradation of only around 5% for most methods. Finally, Figure 3b shows that evaluation metrics are stable against reduced test-set sizes, confirming the feasibility of reliable assessment with limited test data.

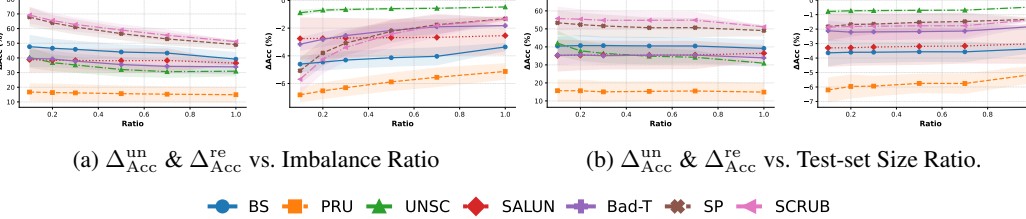

(a) $\Delta_{\mathrm{Acc}}^{\mathrm{un}}$ & $\Delta_{\mathrm{Acc}}^{\mathrm{re}}$ vs. Imbalance Ratio      (b) $\Delta_{\mathrm{Acc}}^{\mathrm{un}}$ & $\Delta_{\mathrm{Acc}}^{\mathrm{re}}$ vs. Test-set Size Ratio.

Figure 3: Impact of data conditions on utility restorability. We measure the accuracy gain on forgotten classes ($\Delta \mathrm{Acc}_{\mathrm{un}}$) to assess knowledge recovery and on remaining classes ($\Delta \mathrm{Acc}_{\mathrm{re}}$) to evaluate collateral impact. The gain is calculated as $\Delta \mathrm{Acc} = \mathrm{Acc}_{\mathrm{restore}} - \mathrm{Acc}_{\mathrm{unlearn}}$, where $\mathrm{Acc}_{\mathrm{unlearn}}$ denotes the model accuracy after unlearning and $\mathrm{Acc}_{\mathrm{restore}}$ denotes its accuracy after restoring the unlearned model's utility.

## 5 CONCLUSION

In this paper, we investigate the illusion of forgetting, the gap between suppressed forgotten-class accuracy and the residual, recoverable information that persists in an unlearned model's outputs. By analyzing the statistical signatures left along unlearning trajectories, we showed that preserving utility on retained classes can inherently leave structured traces about forgotten classes. Building on these insights, we proposed a lightweight, black-box post-hoc recovery procedure that rescales the near-zero outputs via a Yeo–Johnson transformation and applies adaptive thresholding to reconstruct predictions for forgotten classes. Across some unlearning methods and benchmarks, this framework restores forgotten-class utility while minimally affecting performance on the remaining classes. Our findings motivate the development of more robust methods that address recovery risk, such as evaluation-time transformations, output-space regularization, or structural changes that mitigate informative collapse without unduly harming retained utility.

## REPRODUCIBILITY STATEMENT

Upon submission, we will release an anonymized code archive with data prep, training/unlearning wrappers, and our post-hoc recovery implementation on Anonymous Repository. Datasets, backbones, splits, and full hyperparameters are specified in the paper and the Appendix.

## ETHICS STATEMENT

We use only standard public datasets and do not process PII or involve human subjects. Our analysis targets understanding of class unlearning and is confined to a realistic black-box setting (probability outputs only; no access to weights, gradients, or training data). We do not interact with deployed services, and any released artifact is restricted to research models/datasets with risk notes.

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

# APPENDIX

## A  MORE RELATED WORKS

### A.1  MACHINE UNLEARNING

Machine unlearning aims to remove the influence of specific training data from trained models Cao & Yang (2015); Bourtoule et al. (2021); Nguyen et al. (2022). Exact unlearning through retraining from scratch serves as the gold standard but incurs prohibitive computational costs Bourtoule et al. (2021); Kim & Woo (2022). To address this, approximate unlearning methods have emerged across several categories: gradient-based approaches that reverse the learning process Graves et al. (2021); Thudi et al. (2022); Neel et al. (2021), knowledge distillation methods using incompetent teachers Chundawat et al. (2023a); Tarun et al. (2023), model modification techniques including parameter pruning and isolation Jia et al. (2023); Chen et al. (2024); Kurmanji et al. (2023), and boundary manipulation strategies Chen et al. (2023). Recent works also explore orthogonal projected gradient Hoang et al. (2024) and null space calibration approaches Chen et al. (2024). However, these methods primarily focus on achieving low accuracy on forgotten data without considering the recoverability of utility, a critical gap our work addresses.

## A.2 EVALUATION OF MACHINE UNLEARNING

Current evaluation paradigms for machine unlearning primarily rely on performance-based metrics. These approaches measure forgetting quality through accuracy degradation on forgotten data and retention quality through maintained performance on remaining data Graves et al. (2021); Chen et al. (2024). Some work has proposed more sophisticated metrics, such as membership inference attack (MIA) success rate Shokri et al. (2017); Chen et al. (2021), relearn time Chundawat et al. (2023b); Golatkar et al. (2020), and activation pattern analysis Foster et al. (2024).

Recent studies have challenged the sufficiency of accuracy-based evaluation. Information-theory-based approaches attempt to quantify residual information through mutual information bounds Kurmanji et al. (2023), while certification methods provide theoretical guarantees under specific assumptions Guo et al. (2020); Sekhari et al. (2021). Empirical evaluation frameworks have emerged to standardize assessment across different unlearning scenarios Nguyen et al. (2022).

## B DETAILED PROOFS

Here, we provide the complete proof of Theorem 1, Theorem 2, and Theorem 3.

### B.1 PROOF OF THEOREM 1

We establish the lower bound through a series of steps:

**Definition 3** *(Feature Extractor) Let* $h : \mathcal{X} \to \mathbb{R}^m$ *be the feature extractor (all layers except the final). The original model achieves high accuracy by learning discriminative features Bengio et al. (2012):*

$$\min_{h,W} \sum_{i=1}^{n} \ell(W^T h(x_i), y_i). \tag{16}$$

*This creates features where* $h(x)$ *for* $x \in \mathcal{X}_c$ *activates a specific subspace* $V_c$.

**Lemma 4** *(Mutual Information) The mutual information between random variables* $X$ *and* $Y$ *is:*

$$I(X;Y) = H(X) - H(X|Y) = H(Y) - H(Y|X), \tag{17}$$

*where* $H(\cdot)$ *denotes entropy and* $H(\cdot|\cdot)$ *denotes conditional entropy.*

**1) Mutual Information Decomposition.** By the Markov chain $Y_c \to h(X) \to f_{\theta^*}(X)$, where $f_{\theta^*}(X)$ is a deterministic function of $h(X)$, the data processing inequality gives:

$$I(Y_c; f_{\theta^*}(X) \mid X \in \mathcal{X}_c) \le I(Y_c; h(X) \mid X \in \mathcal{X}_c).$$

By the symmetry of mutual information $I(A; B) = I(B; A)$:

$$I(f_{\theta^*}(X); Y_c \mid X \in \mathcal{X}_c) \le I(h(X); Y_c \mid X \in \mathcal{X}_c).$$

Substituting the definition of residual information, i.e., Definition 2:

$$\mathcal{I}_{\text{res}}(c) \le I(h(X); Y_c \mid X \in \mathcal{X}_c).$$

However, to maintain accuracy $\alpha_r \ge \alpha_0$, the features $h(X)$ must retain discriminative information.

**2) Fano's Inequality Application.** For any classifier with error rate $\epsilon = 1 - \alpha_0$ on $K$ classes, based on Fano's Inequality Scarlett & Cevher (2019), we have:

$$H(Y|h(X)) \le H(\epsilon) + \epsilon \log(K - 1), \tag{18}$$

where $H(\epsilon) = -\epsilon \log \epsilon - (1 - \epsilon) \log(1 - \epsilon)$.

**3) Information Lower Bound.** $I(h(X); Y) = H(Y) - H(Y|h(X))$ and $H(Y) = \log K$ for uniform distribution:

$$I(h(X); Y) \ge \log K - H(\epsilon) - \epsilon \log(K - 1). \tag{19}$$

**4) Class-Specific Information.** Due to shared feature learning, each class contributes approximately $1/K$ of the total mutual information. Therefore:

$$I(h(X); Y_c | X \in \mathcal{X}_c) \approx \frac{1}{K} \cdot I(h(X); Y). \tag{20}$$

Combine Eq. 19 with the above equation:

$$I(h(X); Y_c \mid X \in \mathcal{X}_c) \geq \frac{1}{K} \Big[ \log K - H(\epsilon) \\ - \epsilon \log(K-1) \Big]. \tag{21}$$

Substitute error rate $\epsilon = 1 - \alpha_0$ and $H(\epsilon) = -\epsilon \log \epsilon - (1-\epsilon) \log(1-\epsilon)$:

$$\mathcal{I}_{\text{res}}(c) \geq \frac{1}{K} \Big[ \log K - H\left(\frac{1-\alpha_0}{K-1}\right) \\ - (1-\alpha_0) \log(K-1) \Big]. \tag{22}$$

### B.2 A.2 PROOF OF THEOREM 2

We establish that forgetting trajectories for distinct classes necessarily diverge in parameter space.

**1) Gradient Decomposition.** When forgetting class $c$, the unlearning gradient has two components:

$$\nabla_\theta \mathcal{L}_{\text{unlearn}}^{(c)}(\theta) = \nabla_\theta \mathcal{L}_r(\theta; \mathcal{D}_r) + \lambda \nabla_\theta \mathcal{L}_f(\theta; \mathcal{D}_c). \tag{23}$$

The class-specific component is:

$$g_c(\theta) = \mathbb{E}_{x \sim p(x|c)}[\nabla_\theta \ell_f(f_\theta(x), c)], \tag{24}$$

where $\ell_f$ is the forgetting loss.

**2) Gradient Difference.** For distinct classes $c_i, c_j$, we analyze:

$$\|g_{c_i}(\theta) - g_{c_j}(\theta)\|^2 = \Big\| \mathbb{E}_{x \sim p(x|c_i)}[\nabla_\theta \ell_f(f_\theta(x), c_i)] \tag{25}$$

$$- \mathbb{E}_{x \sim p(x|c_j)}[\nabla_\theta \ell_f(f_\theta(x), c_j)] \Big\|^2. \tag{26}$$

Using the fact that the loss function $\ell_f$ is designed to maximize entropy for forgotten classes, we can write:

$$\nabla_\theta \ell_f(f_\theta(x), c) = -\nabla_\theta \log p_\theta(c|x) \cdot \mathbb{I}_{[f_\theta(x) \text{ activates for class } c]}, \tag{27}$$

where $\mathbb{I}[\cdot]$ is the indicator function.

**3) Connection to Feature Activation Patterns.** Let $A_c(\theta) \subset \mathbb{R}^m$ be the set of feature activation patterns for class $c$:

$$A_c(\theta) = \{h_\theta(x) : x \in \mathcal{X}_c\}, \tag{28}$$

where $h_\theta$ is the feature extractor Bengio et al. (2012).

We assume that, for well-trained models, classes have partially disjoint activation regions. The gradient difference satisfies:

$$\|g_{c_i}(\theta) - g_{c_j}(\theta)\|^2 \geq \frac{\lambda^2}{L^2} \cdot V(A_{c_i}(\theta) \triangle A_{c_j}(\theta)), \tag{29}$$

where $L$ is the Lipschitz constant of $\ell_f$, $\triangle$ denotes symmetric difference, and $V(\cdot)$ denotes the volume (Lebesgue measure) in the feature space $\mathbb{R}^m$ Folland (1999).

**4) Volume Lower Bound via Dimensionality.** We derive a lower bound on the volume of non-overlapping activation regions using concentration of measure in high dimensions.

The symmetric difference can be expressed as:

$$V(A_{c_i}(\theta) \triangle A_{c_j}(\theta)) = V(A_{c_i} \setminus A_{c_j}) + V(A_{c_j} \setminus A_{c_i}). \tag{30}$$

For a well-trained model, the activation regions correspond to the support of feature distributions:

$$V(A_{c_i} \setminus A_{c_j}) = \int_{\mathbb{R}^m} \mathbb{P}[h_\theta(X) \in dz | Y = c_i] \cdot \mathbb{I}_{[z \notin A_{c_j}]} dz. \tag{31}$$

In high-dimensional spaces, by the concentration of measure phenomenon Vershynin (2018), feature activations concentrate around class-specific manifolds. For features with effective dimension $d_c$, the measure concentrates in a $d_c$-dimensional subspace of $\mathbb{R}^m$.

The volume of the region where class $c_i$ activates but $c_j$ does not satisfies:

$$V(A_{c_i} \setminus A_{c_j}) \geq \frac{d_{c_i}}{2K} \cdot \mathbb{P}[h_\theta(X) \in A_{c_i} \setminus A_{c_j} | Y = c_i]. \tag{32}$$

The probability that features from class $c_i$ fall outside the activation region of class $c_j$ is bounded by:

$$\mathbb{P}[h_\theta(X) \in A_{c_i} \setminus A_{c_j} | Y = c_i] \geq \beta \cdot \|p(x|c_i) - p(x|c_j)\|_{\text{TV}}, \tag{33}$$

where $\beta > 0$ is a constant depending on the Lipschitz property of $h_\theta$.

By symmetry and combining both terms:

$$\begin{aligned}
V(A_{c_i}(\theta) \triangle A_{c_j}(\theta)) &= V(A_{c_i} \setminus A_{c_j}) + V(A_{c_j} \setminus A_{c_i}) \\
&\geq \frac{d_{c_i}}{2K} \cdot \beta \cdot \|p(x|c_i) - p(x|c_j)\|_{\text{TV}} \\
&\quad + \frac{d_{c_j}}{2K} \cdot \beta \cdot \|p(x|c_i) - p(x|c_j)\|_{\text{TV}} \\
&= \beta \cdot \frac{d_{c_i} + d_{c_j}}{2K} \cdot \|p(x|c_i) - p(x|c_j)\|_{\text{TV}}.
\end{aligned} \tag{34}$$

For strongly separated classes, the volume scales quadratically with the TV distance due to the product structure of the activation regions:

$$\begin{aligned}
V(A_{c_i}(\theta) \triangle A_{c_j}(\theta)) &\geq \kappa \cdot \left( \frac{d_{c_i} + d_{c_j}}{2K} \right) \\
&\quad \cdot \|p(x|c_i) - p(x|c_j)\|_{\text{TV}}^2,
\end{aligned} \tag{35}$$

where $\kappa = \beta^2 / c$ for some constant $c > 0$, and $\|p - q\|_{\text{TV}} = \frac{1}{2} \int |p(x) - q(x)| dx$ is the total variation distance between distributions Villani et al. (2008).

Since $c_i \neq c_j$, we have $\|p(x|c_i) - p(x|c_j)\|_{\text{TV}} \geq \epsilon_0 > 0$ for some constant $\epsilon_0$.

**5) Trajectory Evolution.** Let $\gamma_{c_i}(t)$ and $\gamma_{c_j}(t)$ denote the parameter trajectories when forgetting classes $c_i$ and $c_j$ respectively, starting from $\theta_0$ Li et al. (2018). The trajectories evolve according to:

$$\gamma_{c_i}(t) - \gamma_{c_j}(t) = -\int_0^t \Big[ \nabla_\theta \mathcal{L}_{\text{unlearn}}^{(c_i)}(\gamma_{c_i}(s)) \tag{36}$$

$$-\nabla_\theta \mathcal{L}_{\text{unlearn}}^{(c_j)}(\gamma_{c_j}(s)) \Big] ds. \tag{37}$$

Substituting the gradient decomposition from A.2.1:

$$\|\gamma_{c_i}(t) - \gamma_{c_j}(t)\| \geq \lambda \int_0^t \|g_{c_i}(\gamma_{c_i}(s)) - g_{c_j}(\gamma_{c_j}(s))\| ds \tag{38}$$

$$-\int_0^t \|\nabla_\theta \mathcal{L}_{\text{r}}(\gamma_{c_i}(s)) - O(t^2). \tag{39}$$

**6) Final Lower Bound.** Combining the bounds from Steps 3 and 4:

$$\|g_{c_i}(\theta) - g_{c_j}(\theta)\| \geq \frac{\lambda}{L}\sqrt{c_0 \cdot \frac{d_{c_i} + d_{c_j}}{2K} \cdot \epsilon_0^2}. \tag{40}$$

Substituting this into the trajectory bound from Step 5:

$$\begin{aligned}
\mathbb{E}\big[\|\gamma_{c_i}(t) - \gamma_{c_j}(t)\|_2\big] &\geq \lambda \int_0^t \frac{\lambda}{L}\sqrt{\frac{\kappa\epsilon_0^2(d_{c_i} + d_{c_j})}{2K}}\, ds \\
&\quad - O(t^2) \\
&= \delta(t) \cdot \sqrt{\frac{d_{c_i} + d_{c_j}}{2K}},
\end{aligned} \tag{41}$$

where $\delta(t) = \frac{\lambda^2\kappa\epsilon_0 t}{L\sqrt{2}} - O(t^2)$ is monotonically increasing for small $t$ with $\delta(0) = 0$. The expectation is over randomness in the optimization algorithm (e.g., mini-batch sampling).

### B.3 PROOF OF THEOREM 3

Here, we analyze the recovery bound through information-theoretic analysis of the hypothesis testing problem on model outputs.

**Lemma 5** (*Hoeffding's Inequality for Bernoulli Random Variables Fan et al. (2021)*) *Let* $X_1, X_2, \ldots, X_n$ *be independent Bernoulli random variables, and let* $S_n = X_1 + X_2 + \cdots + X_n$. *Then, for any* $t \geq 0$,

$$\mathbb{P}\left(\left|\overline{X} - \mathbb{E}[\overline{X}]\right| \geq \varepsilon\right) \leq \exp\left(-2n\varepsilon^2\right). \tag{42}$$

We test whether $x$ belongs to the forgotten class $c$. Consider the binary hypothesis test:

$$\begin{aligned}
H_0 &: x \sim p(x|y \neq c); \\
H_1 &: x \sim p(x|y = c).
\end{aligned} \tag{43}$$

Let $Z = f_{\theta^*}(x)_c$ be the test statistic. Under each hypothesis:

$$\begin{aligned}
p_0(z) &= p(Z = z \mid H_0); \\
p_1(z) &= p(Z = z \mid H_1).
\end{aligned} \tag{44}$$

To obtain tractable bounds, we have to define the binary statistic, $B = \mathbb{I}[Z > \tau] = \mathbb{I}[f_{\theta^*}(x)_c > \tau]$. By the data processing inequality Cover (1999):

$$I(B; Y_c) \leq I(Z; Y_c) = I(f_{\theta^*}(X)_c; Y_c) = \mathcal{I}_{\text{res}}(c). \tag{45}$$

This inequality is tight when $\tau$ is chosen optimally.
Let $\tau^*$ be the threshold maximizing $I(B; Y_c)$. Define:

$$\begin{aligned}
q_0 &= P(B = 1 \mid Y \neq c) = P(f_{\theta^*}(X)_c > \tau^* \mid Y \neq c) \\
q_1 &= P(B = 1 \mid Y = c) = P(f_{\theta^*}(X)_c > \tau^* \mid Y = c)
\end{aligned} \tag{46}$$

Without loss of generality, assume $q_1 > q_0$.

For binary variables distribution, with $\pi = P(Y = c) = \frac{1}{K}$, through standard information-theoretic analysis Xu & Raginsky (2017), we can get:

$$I(B; Y_c) \geq 2\pi(1 - \pi)(q_1 - q_0)^2 = \frac{2(K - 1)}{K^2}(q_1 - q_0)^2, \tag{47}$$

where $I(B; Y_c) = H(B) - H(B \mid Y_c)$.

For large $K$, we can get:

$$(q_1 - q_0)^2 \geq \frac{K}{2} \cdot I(B; Y_c) \geq \frac{K}{2} \cdot \mathcal{I}_{\text{res}}(c) \tag{48}$$

Given $n$ test samples $\{x_1, \ldots, x_n\}$, and $n$ is finite, we obtain:

$$\hat{B}_n = \frac{1}{n} \sum_{i=1}^{n} \mathbb{I}_{[f_{\theta^*}(x_i)_c > \tau^*]} \tag{49}$$

This is the empirical fraction of samples with logit scores exceeding $\tau^*$. Under $H_0$ (majority of samples from non-class $c$): $\mathbb{E}[\hat{B}_n] \approx q_0$; under $H_1$ (majority of samples from class $c$): $\mathbb{E}[\hat{B}_n] \approx q_1$. Since $\mathbb{I}[f_{\theta^*}(x_i)_c > \tau^*] \in \{0, 1\}$, by Hoeffding's inequality Fan et al. (2021):

$$P\left(\hat{B}_n \geq q_0 + t \mid H_0\right) \leq \exp(-2nt^2) \tag{50}$$

$$P\left(\hat{B}_n \leq q_1 - t \mid H_1\right) \leq \exp(-2nt^2) \tag{51}$$

Substitute threshold $\eta = \frac{q_0 + q_1}{2}$. Setting $t = \frac{q_1 - q_0}{2}$:

$$P(\text{Type I error}) = P\left(\hat{B}_n > \eta \mid H_0\right)$$
$$\leq \exp\left(-\frac{n(q_1 - q_0)^2}{2}\right) \tag{52}$$

$$P(\text{Type II error}) = P\left(\hat{B}_n \leq \eta \mid H_1\right)$$
$$\leq \exp\left(-\frac{n(q_1 - q_0)^2}{2}\right) \tag{53}$$

Using the decision threshold $\eta = \frac{q_0 + q_1}{2}$, we can get the overall error probability:

$$P(\text{error}) \leq \exp\left(-\frac{n(q_1 - q_0)^2}{2}\right) \tag{54}$$

Substituting the bound from Eq. 47:

$$P(\text{error}) \leq \exp\left(-\frac{nK\mathcal{I}_{\text{res}}(c)}{4}\right) \tag{55}$$

For typical machine learning settings with moderate $K$ and considering implementation factors (suboptimal threshold, finite sample effects), we obtain the conservative bound:

$$P[\text{recovery}] = 1 - P(\text{error}) \geq 1 - \exp(-2n \cdot \mathcal{I}_{\text{res}}(c)) \tag{56}$$

where the constant 2 absorbs the dependence on $K$ and other factors, providing a practical bound that holds across diverse settings.

## C  THE ALGORITHM OF THE RECOVERY FRAMEWORK

## D  ADDITIONAL EXPERIMENTAL DETAILS

### D.1  ADDITIONAL TRAINING SETTINGS

We summarize the original model training settings in Tab. 3.

### D.2  CONSIDERED MU METHODS AND THEIR IMPLEMENTATION DETAILS

This section provides the details of the considered MU methods and their implementation details. In the following tables, un-LR denotes the learning rate for unlearning, and un-Epochs denotes the number of epochs for unlearning. There are two settings for each method, corresponding to single-class and multi-class unlearning settings (single/multi).

---

**Algorithm 1** Post-hoc Utility Recovery Framework

---

**Require:** Unlearned model $f_{\theta^*}$, test set $\mathcal{D}_{test}$
**Ensure:** Recovered predictions $\hat{y}$
1: # **Forgotten Class Identification**
2: **for** $k \in \{1, ..., K\}$ **do**
3:     $\mathcal{P}_k \leftarrow \{p_k(x) : x \in \mathcal{D}_{test}\}$ where $p_k(x) = S(f_{\theta^*}(x))_k$
4: **end for**
5: $\mathcal{C}_f \leftarrow \{k : \bar{p}_k < 0.1/K\}$
6: # **Probability Distribution Extraction & Re-scaling**
7: $\mathcal{P}_c \leftarrow \{\mathcal{P}_i\}_{i \in C_f}$
8: $\lambda^* \leftarrow \arg\max_\lambda \sum_{p \in \mathcal{P}_c} \log p_\lambda(p)$
9: $\tilde{\mathcal{P}}_c \leftarrow \{(\mathcal{T}_{\lambda^*}(p)) : p \in \mathcal{P}\}$
10: # **Adaptive Threshold Determination**
11: $\tau^* \leftarrow$ Otsu'$(\{\tilde{p} : \tilde{p} \in \tilde{\mathcal{P}}\})$
12: # **Re-prediction**
13: **for** $x \in \mathcal{D}_{test}$ **do**
14:     $p \leftarrow S(f_{\theta^*}(x))_c$
15:     $\tilde{p} \leftarrow \mathcal{T}_{\lambda^*}(p)$
16:     **if** $\tilde{p} > \tau^*$ **then**
17:         $\hat{y} \leftarrow c$
18:     **else**
19:         $\hat{y} \leftarrow \arg\max_{j \neq c} S(f_{\theta^*}(x))_j$
20:     **end if**
21: **end for**
22:
23: **return** $\{\hat{y}\}$

---

Table 3: Original model training settings.

| Settings | MNIST | FMNIST | CIFAR-10 | CIFAR-100 |
|---|---|---|---|---|
| | AllCNN | AllCNN | ResNet-18 | ResNet-34 |
| Batch Size | 128 | 128 | 128 | 128 |
| Learning Rate | 0.01 | 0.01 | 0.01 | 0.01 |
| Epochs | 30 | 60 | 120 | 200 |
| Optimizer | SGD | SGD | SGD | SGD |
| Weight Decay | $1e^{-4}$ | $1e^{-4}$ | $5e^{-4}$ | $1e^{-4}$ |
| Momentum | 0.9 | 0.9 | 0.9 | 0.9 |
| Scheduler | - | MultiStepLR | MultiStepLR | MultiStepLR |
| Milestones | - | [25, 45] | [60,90] | [100, 150] |

*Gradient-based Methods:*
**Unrolling** Thudi et al. (2022): Approximates retraining by unrolling SGD steps.

Table 4: Unrolling settings.

| Settings | MNIST | FMNIST | CIFAR-10 | CIFAR-100 |
|---|---|---|---|---|
| un-LR | 0.30/0.31 | 0.22/0.17 | 0.20/0.017 | 0.02/0.01 |
| Sigma | 0.036/0.011 | 0.028/0.01 | 0.010/0.010 | 0.020/0.025 |

**Unrolling-F**: Unrolling is applied only to forgetting samples.

Table 5: Unrolling-F settings.

| Settings | MNIST | FMNIST | CIFAR-10 | CIFAR-100 |
|---|---|---|---|---|
| un-LR | 0.0040/0.0040 | 0.0040/0.0018 | 0.050/0.016 | 0.0080/0.0024 |
| Sigma | 0.0035/0.0036 | 0.0015/0.0042 | 0.040/0.015 | 0.030/0.022 |

**GA (Gradient Ascent)** Golatkar et al. (2020): Maximizes loss on forgetting data.

Table 6: GA settings.

| Settings | MNIST | FMNIST | CIFAR-10 | CIFAR-100 |
|---|---|---|---|---|
| un-LR | 4.7e-5/5.1e-5 | 8.0e-7/2.8e-4 | 5.8e-4/4.6e-4 | 5.0e-4/1.8e-6 |
| un-Epochs | 5/5 | 40/5 | 5/5 | 10/20 |

**Fisher** Golatkar et al. (2020): Uses Fisher information matrix for selective forgetting.

Table 7: Fisher settings.

| Settings | MNIST | FMNIST | CIFAR-10 | CIFAR-100 |
|---|---|---|---|---|
| alpha | 1e-7/1e-7 | 1e-7/1e-7 | 1e-8/1e-8 | 1e-8/1e-8 |
| un-Epochs | 3/3 | 3/3 | 3/3 | 3/3 |

*Boundary Manipulation:*
**Boundary Shrink** Chen et al. (2023): Contracts decision boundaries around forgotten class.
**Boundary Expand** Chen et al. (2023): Expands boundaries to exclude forgotten class.

Table 8: Boundary Shrink and Boundary Expand settings.

| Settings | MNIST | FMNIST | CIFAR-10 | CIFAR-100 |
|---|---|---|---|---|
| un-LR (Boundary Shrink) | 4.3e-5/5.0e-5 | 1.0e-5/5.2e-4 | 3.3e-4/3.0e-4 | 5.0e-4/1.7e-4 |
| un-LR (Boundary Expand) | 3.6e-5/3.6e-5 | 5.0e-5/1.63e-5 | 5.0e-5/1.14e-5 | 1.0e-5/1.0e-5 |

*Knowledge Distillation:* **Bad Teacher** Chundawat et al. (2023a): Uses incompetent teacher for selective forgetting.

Table 9: Bad Teacher settings.

| Settings | MNIST | FMNIST | CIFAR-10 | CIFAR-100 |
|---|---|---|---|---|
| un-LR | 7.0e-3/1.0e-2 | 5.0e-2/4.0e-2 | 1.0e-1/9.2e-2 | 7.5e-2/9.4e-3 |
| un-Epochs | 7/3 | 5/5 | 60/35 | 40/40 |
| Temperature | 1.0/1.0 | 1.0/1.0 | 1.2/2.3 | 2.7/2.5 |

*Model Modification:* **Sparse Unlearning** Jia et al. (2023): Leverages model sparsity for efficient unlearning.

Table 10: Sparse Unlearning settings.

| Settings | MNIST | FMNIST | CIFAR-10 | CIFAR-100 |
|---|---|---|---|---|
| un-LR | 0.01/0.01 | 0.01/0.01 | 0.01/0.01 | 0.01/0.01 |
| un-Epochs | 10/10 | 10/10 | 10/10 | 10/10 |
| Pruning-Rate | 0.95/0.95 | 0.95/0.95 | 0.95/0.95 | 0.95/0.95 |

**SCRUB** Kurmanji et al. (2023): Selective gradient updates with regularization.

Table 11: SCRUB settings.

| Settings | MNIST | FMNIST | CIFAR-10 | CIFAR-100 |
|---|---|---|---|---|
| un-LR | 2.7e-6/2.0e-5 | 2.0e-5/2.0e-5 | 1e-5/1e-5 | 4.4e-6/4.4e-6 |
| un-Epochs | 3/3 | 3/3 | 3/3 | 35/35 |
| alpha | 0.57/0.30 | 0.64/0.30 | 0.57/0.60 | 0.51/0.41 |
| gamma | 2.70/1.00 | 1.00/1.00 | 1.00/1.00 | 4.78/4.97 |

**SALUN** Fan et al. (2024): Gradient-based weight saliency approach.

Table 12: SALUN settings.

| Settings | MNIST | FMNIST | CIFAR-10 | CIFAR-100 |
|---|---|---|---|---|
| un-LR | 3.7e-5/2.0e-5 | 4.0e-5/7.0e-5 | 4.5e-4/4.4e-5 | 5.5e-4/5.6e-4 |
| un-Epochs | 15/10 | 8/8 | 15/15 | 7/7 |
| threshold | 0.57/0.7 | 0.64/0.3 | 0.57/0.57 | 0.51/0.54 |

*Null Space Methods:* **UNSC** Chen et al. (2024): Projects gradients onto null space.

Table 13: UNSC settings.

| Settings | MNIST | FMNIST | CIFAR-10 | CIFAR-100 |
|---|---|---|---|---|
| un-LR | 0.01/0.01 | 0.01/0.01 | 0.05/0.05 | 0.05/0.05 |
| un-Epochs | 10/10 | 15/15 | 25/25 | 25/25 |

**PRU** Zhang et al. (2024): Perception revising unlearning.

Table 14: PRU settings.

| Settings | MNIST | FMNIST | CIFAR-10 | CIFAR-100 |
|---|---|---|---|---|
| shift epoch | 15/10 | 10/10 | 10/10 | 50/10 |
| shift lr | 2.8e-5/8.0e-5 | 1.0e-5/2.0e-5 | 1.0e-4/5.0e-4 | 1.35e-6/2.00e-6 |
| shift lambda | 1 | 1 | 1 | 1 |
| refine epochs | 5/1 | 1/1 | 2/1 | 40/1 |
| refine lr | 1.45e-3/3.00e-3 | 2.00e-3/1.00e-3 | 4.00e-3/5.00e-3 | 2.00e-2/2.00e-2 |

### D.3 DETAILS OF THE OOD DETECTION METHODS

**Max Softmax Probability (MSP) Hendrycks & Gimpel (2017).** Detect OOD by thresholding the model's maximum predicted class probability. Lower MSP indicates higher OOD likelihood. The MSP score is defined as:

$$\mathrm{MSP}(x) = \max_{k \in \{1,\ldots,K\}} p_k(x), \qquad p_k(x) = \frac{e^{f_k(x)}}{\sum_{j=1}^{K} e^{f_j(x)}}.$$

**Entropy-based detection Malinin & Gales (2018).** Use predictive uncertainty via Shannon entropy; higher entropy suggests OOD. The entropy score is defined as:

$$H(x) = -\sum_{k=1}^{K} p_k(x) \log p_k(x).$$

Decide OOD by thresholding $H(x)$ (or equivalently use $-H(x)$ as an in-distribution score).

**Gini (impurity) score Liu et al. (2023).** Measure concentration of the predictive distribution using Gini impurity. A higher Gini score indicates a more uniform distribution, which suggests OOD. The Gini score is defined as:

$$\mathrm{Gini}(x) = 1 - \sum_{k=1}^{K} p_k(x)^2.$$

**Ratio (likelihood ratio) Ren et al. (2019).** Correct raw likelihoods using a background model to discount generic statistics. Lower ratio indicates OOD. The ratio score is defined as:

$$s_{\mathrm{LR}}(x) = \log p_\theta(x) - \log p_{\mathrm{bg}}(x) = \log \frac{p_\theta(x)}{p_{\mathrm{bg}}(x)}.$$

**Energy Score (overall-based) Liu et al. (2020).** Compute the (free) energy from logits via log-sum-exp; higher energy indicates OOD. The energy score is defined as:

$$E(x; f) = T \log \sum_{k=1}^{K} \exp\left(\frac{f_k(x)}{T}\right).$$

Often, the negative energy $-E(x; f)$ is used as an in-distribution score.

**Energy Score (label-/class-specific; joint) Wang et al. (2021).** For multi-label settings, aggregate label-wise energies to capture joint evidence. Higher joint energy suggests OOD. The joint energy score is defined as:

$$E_k(x) = \log\big(1 + e^{f_k(x)}\big), \qquad E_{\text{joint}}(x) = \sum_{k=1}^{K} E_k(x).$$

where $E_k(x)$ is the energy score for class $k$. Threshold $-E_{\text{joint}}(x)$ as an in-distribution score.

In the summarization of the above methods, $f_k(x)$ denotes the logits for class $k$, $p_k(x) = \frac{e^{f_k(x)}}{\sum_j e^{f_j(x)}}$, $K$ is the number of classes, and $T > 0$ is a temperature.

# E  ADDITIONAL EXPERIMENTAL RESULTS

## E.1  RECOVERY RESULTS FOR SINGLE-CLASS UNLEARNING

Table 15: **Results of Unlearning and Recovery on Single-Class forgetting** (mean $\pm$ std over three runs). Upper line: original unlearned results; lower line (with gray background): post-hoc recovered results. FA is accuracy on forgotten classes; RA is accuracy on remaining classes.

| Method | MNIST | | FMNIST | | CIFAR-10 | | CIFAR-100 | |
|---|---|---|---|---|---|---|---|---|
| | FA | RA | FA | RA | FA | RA | FA | RA |
| Orig | 99.54±0.26 | 99.55±0.03 | 93.00±6.69 | 93.00±0.74 | 92.96±3.94 | 92.96±0.44 | 74.63±11.21 | 72.09±0.11 |
| RT | 0.00±0.00 | 99.49±0.56 | 0.00±0.00 | 93.60±1.23 | 0.00±0.00 | 93.32±0.65 | 0.00±0.00 | 72.06±0.32 |
| | 89.53±8.70 | 97.43±1.87 | 49.25±14.58 | 87.87±0.90 | 44.67±10.62 | 87.28±1.12 | 46.57±12.38 | 67.77±0.65 |
| PRU | 0.00±0.00 | 98.19±1.68 | 0.19±0.27 | 91.45±3.86 | 0.00±0.00 | 93.06±0.79 | 0.00±0.00 | 70.61±1.29 |
| | 90.87±3.13 | 98.18±1.69 | 84.41±12.99 | 90.77±4.71 | 88.34±5.15 | 92.13±0.58 | 81.13±8.84 | 70.07±1.38 |
| BS | 10.60±6.72 | 95.58±4.68 | 0.08±0.12 | 91.04±1.69 | 6.56±1.34 | 91.34±0.89 | 0.53±0.78 | 59.34±3.44 |
| | 90.77±4.97 | 95.40±5.00 | 77.62±12.92 | 90.76±1.60 | 63.37±6.47 | 91.27±0.84 | 52.77±9.31 | 59.16±3.44 |
| BE | 12.57±9.27 | 83.26±10.00 | 11.80±3.16 | 89.00±5.83 | 24.75±4.42 | 91.43±1.15 | 2.17±1.74 | 62.26±4.66 |
| | 89.20±4.80 | 82.33±10.98 | 89.58±9.42 | 87.88±7.14 | 83.79±5.90 | 91.11±1.03 | 50.03±9.73 | 62.20±4.67 |
| Unroll | 0.00±0.00 | 87.45±16.66 | 0.03±0.10 | 82.15±13.66 | 5.45±5.72 | 90.17±6.14 | 12.73±10.01 | 71.32±0.36 |
| | 80.22±21.24 | 80.27±18.19 | 69.14±17.49 | 78.80±14.33 | 77.22±11.48 | 89.39±6.34 | 66.80±12.72 | 71.24±0.35 |
| Unroll-F | 0.24±0.55 | 94.97±5.01 | 59.53±13.26 | 90.75±3.43 | 3.40±3.73 | 88.66±1.92 | 2.33±3.48 | 70.03±0.60 |
| | 95.94±2.62 | 93.00±9.52 | 88.06±9.88 | 90.41±3.47 | 47.79±18.03 | 88.63±1.91 | 38.67±9.85 | 70.03±0.60 |
| UNSC | 0.02±0.04 | 99.49±0.07 | 0.16±0.24 | 93.47±1.16 | 0.00±0.00 | 93.27±0.71 | 0.00±0.00 | 71.73±0.32 |
| | 83.99±4.85 | 98.48±1.02 | 61.99±20.07 | 89.64±2.01 | 40.86±9.14 | 92.05±0.78 | 45.07±18.14 | 70.96±0.77 |
| SALUN | 0.16±0.28 | 98.28±1.40 | 0.64±0.44 | 91.83±1.02 | 1.84±1.71 | 88.12±1.49 | 3.60±3.39 | 66.98±1.38 |
| | 96.94±1.23 | 98.25±1.40 | 88.67±8.58 | 91.43±0.99 | 84.20±13.07 | 87.12±1.59 | 80.60±10.40 | 66.25±1.68 |
| GA | 9.62±3.85 | 93.00±8.69 | 12.39±4.02 | 89.68±4.47 | 4.38±2.11 | 84.99±2.88 | 1.03±1.35 | 63.42±4.06 |
| | 84.04±6.60 | 92.53±9.97 | 83.39±10.48 | 89.36±4.63 | 29.03±2.70 | 84.98±2.88 | 41.83±10.96 | 63.39±4.07 |
| Fisher | 2.63±5.31 | 98.87±0.51 | 3.67±10.05 | 92.18±1.34 | 5.06±5.89 | 92.63±0.75 | 0.00±0.00 | 70.30±0.26 |
| | 99.11±0.60 | 95.62±2.81 | 87.25±9.99 | 89.47±1.25 | 85.67±5.04 | 90.77±0.70 | 54.57±13.65 | 65.80±0.32 |
| Bad-T | 0.00±0.00 | 98.72±0.40 | 1.02±2.67 | 90.70±1.56 | 1.64±2.96 | 91.96±1.46 | 1.37±1.94 | 66.50±0.61 |
| | 99.90±0.10 | 94.79±1.98 | 97.98±1.85 | 86.85±2.24 | 96.26±2.34 | 90.00±1.23 | 83.50±9.10 | 65.68±0.65 |
| SP | 0.00±0.00 | 99.47±0.05 | 0.00±0.00 | 92.69±1.41 | 0.00±0.00 | 92.50±0.74 | 0.83±1.84 | 70.68±0.37 |
| | 95.32±3.82 | 98.53±0.34 | 74.36±11.43 | 85.31±2.36 | 67.05±8.19 | 86.65±1.20 | 84.77±4.86 | 69.44±0.79 |
| SCRUB | 0.36±1.17 | 99.29±0.08 | 2.44±7.28 | 91.49±1.53 | 0.01±0.04 | 89.33±0.96 | 7.57±8.67 | 66.22±0.33 |
| | 97.02±1.63 | 99.09±0.14 | 88.83±6.59 | 89.73±1.90 | 89.98±3.84 | 85.92±1.99 | 88.90±4.30 | 64.79±0.79 |

## E.2  THE COMPLETE RESULTS OF COMPARISON METHODS ON CIFAR-100

These are complementary results for Tab. 2; we additionally include RT and Fisher.

Table 16: Comparing the Recall among the proposed method and OOD detection methods on **CIFAR-100**.

| Method | RT | PRU | BS | BE | Unroll | Unroll-F | UNSC | SALUN | GA | Fisher | Bad-T | SP | SCRUB |
|---|---|---|---|---|---|---|---|---|---|---|---|---|---|
| MSP | 1.67±0.39 | 3.92±1.39 | 7.86±1.59 | 15.44±0.39 | 11.65±0.70 | 13.46±0.67 | 15.30±1.18 | 2.39±1.06 | 11.16±0.62 | 1.83±0.59 | 10.49±1.05 | 13.74±0.86 | 12.40±0.65 |
| Ratio | 1.66±0.40 | 4.11±1.41 | 7.76±1.50 | 14.61±0.49 | 11.48±0.84 | 12.98±0.85 | 14.79±1.12 | 2.44±1.16 | 10.74±0.75 | 1.82±0.58 | 9.73±1.17 | 14.40±1.01 | 13.52±0.81 |
| Energy | 1.26±0.30 | 3.16±1.10 | 6.45±1.25 | 12.68±0.54 | 10.94±0.80 | 11.87±0.73 | 12.13±0.79 | 2.24±1.14 | 11.12±0.76 | 1.38±0.46 | 8.32±0.78 | 11.33±0.67 | 10.74±0.69 |
| Margin | 1.62±0.40 | 3.77±1.33 | 7.50±1.51 | 14.83±0.44 | 11.50±0.69 | 13.05±0.63 | 14.89±1.11 | 2.37±1.04 | 10.96±0.65 | 1.75±0.56 | 9.68±0.97 | 13.25±0.83 | 11.99±0.61 |
| Gini | 1.58±0.37 | 3.57±1.28 | 7.34±1.46 | 14.82±0.42 | 11.53±0.63 | 13.05±0.57 | 14.86±1.00 | 2.35±0.99 | 11.07±0.59 | 1.69±0.54 | 9.50±0.93 | 12.66±0.72 | 11.40±0.53 |
| Energy+ | 1.48±0.31 | 3.38±1.22 | 7.29±1.40 | 15.27±0.36 | 11.70±0.53 | 13.47±0.48 | 15.29±1.13 | 2.36±1.03 | 11.80±0.49 | 1.62±0.53 | 10.29±1.00 | 12.38±0.63 | 11.30±0.58 |
| Entropy | 1.72±0.38 | 4.04±1.43 | 8.45±1.72 | 16.81±0.40 | 11.89±0.78 | 14.30±0.74 | 16.40±1.32 | 2.37±1.05 | 11.73±0.69 | 1.93±0.63 | 12.98±1.24 | 14.26±0.95 | 12.55±0.74 |
| **Ours** | **1.99±0.39** | **7.80±3.31** | **14.46±5.31** | **94.74±1.89** | **64.23±4.76** | **71.04±1.47** | **66.03±7.13** | **34.42±1.41** | **19.61±1.15** | **2.50±0.88** | **29.60±6.54** | **58.55±1.14** | **62.38±1.06** |

### E.3  MIA EVALUATION

Table 17: **Membership inference (MIA) for Single-class unlearning.** $\text{AUROC}_F/\text{AUROC}_R$ are AUCs on the forgotten/retained subsets (positive = member). F-TPR5/R-TPR5 denote the true positive rate at FPR = 5% on the forgotten/retained subsets. $\mathbf{MIA_I}$ (SVM–TNR) is the true-negative rate on the forgotten set, $TN/|D_u|$ (↑ indicates stronger privacy). $\mathbf{MIA_{II}}$ is the attacker accuracy at the shadow-optimal threshold $\tau^*$ chosen to maximize balanced shadow accuracy (values near 50% indicate chance).

| Method | MNIST | | | | | | Fashion-MNIST | | | | | |
|---|---|---|---|---|---|---|---|---|---|---|---|---|
| | $\text{AUROC}_F$ | $\text{AUROC}_R$ | $F-\text{TPR5}$ | $R-\text{TPR5}$ | $\text{MIA}_I$ | $\text{MIA}_{II}$ | $\text{AUROC}_F$ | $\text{AUROC}_R$ | $F-\text{TPR5}$ | $R-\text{TPR5}$ | $\text{MIA}_I$ | $\text{MIA}_{II}$ |
| RT | 49.93±0.09 | 50.01±0.04 | 4.88±0.30 | 4.57±0.09 | 100.00±0.00 | 51.89±0.49 | 49.97±0.09 | 49.25±0.31 | 4.87±0.26 | 2.56±0.73 | 100.00±0.00 | 58.50±1.99 |
| PRU | 50.01±0.09 | 49.95±0.06 | 4.95±0.40 | 4.64±0.16 | 99.99±0.03 | 50.81±0.38 | 49.56±0.27 | 49.22±0.30 | 3.49±0.63 | 2.80±0.71 | 100.00±0.00 | 57.01±1.57 |
| BS | 49.98±0.14 | 49.94±0.10 | 4.89±0.41 | 4.31±0.79 | 94.97±13.68 | 50.74±0.30 | 49.61±0.32 | 48.95±0.40 | 3.92±0.49 | 2.15±0.81 | 99.47±1.60 | 55.25±1.30 |
| BE | 49.91±0.15 | 49.91±0.08 | 4.52±0.40 | 4.02±0.84 | 92.08±10.61 | 50.52±0.33 | 48.90±0.77 | 49.21±0.35 | 2.98±0.81 | 1.93±0.64 | 99.37±0.40 | 55.73±0.73 |
| Unroll | 50.01±0.17 | 49.99±0.06 | 5.06±0.48 | 4.78±0.46 | 58.29±30.03 | 50.35±0.33 | 49.90±0.16 | 49.68±0.26 | 4.64±0.90 | 4.12±0.85 | 80.27±30.57 | 50.90±0.89 |
| Unroll-F | 50.01±0.12 | 49.89±0.05 | 4.92±0.38 | 4.26±0.48 | 88.47±16.54 | 50.47±0.24 | 49.61±0.37 | 48.98±0.36 | 4.27±0.96 | 1.07±0.77 | 98.90±2.15 | 53.96±1.08 |
| UNSC | 50.33±0.16 | 49.94±0.04 | 4.94±0.29 | 1.61±0.21 | 100.00±0.00 | 51.22±0.23 | 49.88±0.26 | 49.18±0.24 | 3.70±0.61 | 0.27±0.97 | 100.00±0.00 | 57.06±2.07 |
| SALUN | 50.04±0.12 | 50.01±0.04 | 4.94±0.29 | 4.78±0.10 | 100.00±0.00 | 50.97±0.27 | 49.62±0.16 | 49.68±0.32 | 3.33±0.47 | 2.19±0.55 | 100.00±0.00 | 56.02±1.44 |
| GA | 49.90±0.45 | 49.92±0.08 | 4.95±0.29 | 2.00±1.84 | 84.40±18.17 | 50.49±0.26 | 49.60±0.47 | 49.08±0.31 | 4.50±0.44 | 1.40±1.30 | 97.86±3.48 | 53.48±0.88 |
| Fisher | 49.98±0.10 | 49.97±0.04 | 4.96±0.28 | 4.65±0.13 | 95.06±12.57 | 50.75±0.25 | 49.43±0.29 | 49.13±0.35 | 3.67±0.54 | 2.48±0.79 | 100.00±0.00 | 55.69±1.17 |
| Bad-T | 50.01±0.13 | 50.00±0.05 | 5.12±0.39 | 4.79±0.14 | 36.85±37.02 | 50.25±0.38 | 49.82±0.09 | 49.66±0.18 | 4.30±0.35 | 3.62±0.42 | 99.96±0.12 | 51.85±0.31 |
| SP | 49.93±0.10 | 49.99±0.02 | 4.76±0.26 | 4.57±0.11 | 99.89±0.52 | 50.88±0.16 | 49.93±0.05 | 49.11±0.32 | 4.80±0.18 | 2.48±0.74 | 100.00±0.00 | 55.65±1.29 |
| SCRUB | 49.94±0.12 | 50.02±0.04 | 4.91±0.37 | 4.77±0.16 | 67.34±24.97 | 50.48±0.44 | 49.84±0.08 | 49.43±0.25 | 4.54±0.20 | 3.37±0.66 | 98.48±4.20 | 52.12±0.89 |

| M | CIFAR-10 | | | | | | CIFAR-100 | | | | | |
|---|---|---|---|---|---|---|---|---|---|---|---|---|
| | $\text{AUROC}_F$ | $\text{AUROC}_R$ | $F-\text{TPR5}$ | $R-\text{TPR5}$ | $\text{MIA}_I$ | $\text{MIA}_{II}$ | $\text{AUROC}_F$ | $\text{AUROC}_R$ | $F-\text{TPR5}$ | $R-\text{TPR5}$ | $\text{MIA}_I$ | $\text{MIA}_{II}$ |
| RT | 50.02±0.13 | 49.21±0.15 | 5.14±0.32 | 0.20±0.76 | 100.00±0.00 | 57.46±0.93 | 50.09±0.18 | 47.30±0.12 | 5.21±0.35 | 0.00±0.00 | 100.00±0.00 | 75.86±0.59 |
| PRU | 49.70±0.13 | 49.38±0.14 | 3.89±0.44 | 2.83±0.45 | 100.00±0.00 | 57.00±0.80 | 49.92±0.09 | 48.21±0.16 | 4.56±0.40 | 0.02±0.01 | 100.00±0.00 | 72.45±0.58 |
| BS | 49.68±0.11 | 49.02±0.15 | 4.14±0.26 | 1.44±0.64 | 99.87±0.50 | 54.44±0.74 | 48.62±0.36 | 45.82±0.18 | 3.61±0.36 | 0.92±0.21 | 97.96±0.92 | 67.17±1.18 |
| BE | 46.92±0.97 | 49.09±0.17 | 2.54±0.69 | 0.03±0.02 | 92.64±2.71 | 56.71±0.75 | 41.65±0.89 | 46.93±0.12 | 0.34±0.14 | 0.01±0.00 | 72.44±2.73 | 74.31±0.55 |
| Unroll | 46.94±1.09 | 49.13±0.20 | 2.83±0.80 | 0.49±0.34 | 82.86±5.59 | 56.42±0.81 | 44.40±1.09 | 47.19±0.15 | 0.41±0.43 | 0.00±0.00 | 36.32±10.00 | 75.00±0.76 |
| Unroll-F | 48.72±0.96 | 48.97±0.21 | 3.77±0.63 | 0.97±0.67 | 92.73±3.39 | 55.25±1.21 | 43.10±0.84 | 46.84±0.23 | 0.75±0.60 | 0.02±0.02 | 56.96±10.25 | 73.61±1.02 |
| UNSC | 49.73±0.15 | 48.99±0.18 | 4.11±0.43 | 0.00±0.00 | 100.00±0.00 | 56.01±0.67 | 49.60±0.17 | 46.66±0.12 | 4.02±0.50 | 0.00±0.00 | 100.00±0.00 | 73.70±0.50 |
| SALUN | 49.80±0.10 | 49.44±0.11 | 4.41±0.33 | 2.23±0.92 | 99.98±0.04 | 53.40±0.86 | 49.96±0.10 | 49.89±0.36 | 4.78±0.43 | 4.78±0.31 | 76.35±12.06 | 50.90±1.82 |
| GA | 49.87±0.09 | 48.97±0.14 | 4.82±0.25 | 1.81±0.70 | 99.48±0.92 | 53.68±0.80 | 45.36±0.76 | 47.08±0.10 | 0.02±0.01 | 0.00±0.00 | 18.10±3.37 | 74.89±0.52 |
| Fisher | 49.30±0.25 | 49.26±0.14 | 3.27±0.43 | 2.50±0.36 | 100.00±0.00 | 56.90±0.74 | 49.45±0.11 | 47.24±0.12 | 3.30±0.47 | 0.01±0.00 | 100.00±0.00 | 74.28±0.50 |
| Bad-T | 49.40±0.18 | 49.24±0.26 | 3.05±0.56 | 1.69±0.97 | 100.00±0.00 | 55.44±0.61 | 49.63±0.10 | 47.92±0.10 | 4.08±0.34 | 3.00±0.13 | 97.89±4.47 | 56.97±0.44 |
| SP | 49.88±0.07 | 49.21±0.15 | 4.63±0.25 | 2.28±0.61 | 100.00±0.00 | 56.24±0.80 | 49.79±0.07 | 47.79±0.10 | 4.33±0.28 | 0.21±0.02 | 99.98±0.04 | 66.56±0.46 |
| SCRUB | 49.87±0.08 | 49.34±0.14 | 4.61±0.32 | 3.01±0.37 | 99.93±0.27 | 53.12±0.58 | 49.66±0.13 | 47.50±0.12 | 4.19±0.32 | 0.89±0.06 | 95.25±2.46 | 62.43±0.55 |

**Membership inference evaluation.** We evaluate *MIA* on the unlearned model to quantify whether training membership can be inferred from outputs Shokri et al. (2017); Jia et al. (2023); Chen et al. (2024); Kurmanji et al. (2023); Chen et al. (2023). To stabilize the attack across datasets and methods, we feed a *fusion feature* that is the concatenation of five prediction statistics into a binary SVM, including *correctness* (argmax equals the label), *confidence* $p_y$, *entropy* $H(p)$, *loss* $-\log p_y$, and *margin* $p_{(1)} - p_{(2)}$. MIA proceeds in two phases: (1) a shadow training phase where we firstly construct a balanced retained shadow dataset (retained training data as members, retained test data as non-members) and learn the SVM decision; and (2) an attack phase where the learned attacker is applied to the forgotten set $D_u$ and retained evaluation splits. We report two metrics following the existing works: $\mathbf{MIA_I}$ is the true negative rate on the forgotten set, $\text{TN}/|D_u|$ (higher ↑ indicates stronger privacy); and $\mathbf{MIA_{II}}$ is the attacker's accuracy at the shadow-optimal threshold $\tau^*$ chosen to maximize balanced shadow accuracy (values near 50% indicate chance). In addition, we include AUROC and TPR@5%FPR as complementary, widely used summaries. AUROC is threshold-free and prevalence-robust, which equals the probability that a random member receives a higher attack score than a random non-member (0.5 = chance). TPR@5%FPR captures operational attack strength under a low false-positive budget that is relevant for privacy claims. Concretely, for each example $x$ we compute an attack score $s(x)$ (SVM decision value or one of the scalar scores), form member/non-member score sets for the subset of interest (forgotten $F$ or retained $R$), compute AUROC from the ROC curve, and obtain TPR@5%FPR by selecting a threshold on a *retained-only, balanced* validation set to achieve FPR $\approx 0.05$ (no test-label peeking), then

measuring the corresponding TPR on the evaluation split. All operating-point metrics are computed with balanced sampling, and results are averaged over multiple seeds for statistical stability.

Table 18: **Membership inference (MIA) for Multi-class unlearning.** $AUROC_F$/$AUROC_R$ are AUCs on the forgotten/retained subsets (positive = member). F-TPR5/R-TPR5 denote the true positive rate at $FPR = 5\%$ on the forgotten/retained subsets. $MIA_I$ (SVM–TNR) is the true-negative rate on the forgotten set, $TN/|D_u|$ ($\uparrow$ indicates stronger privacy). $MIA_{II}$ is the attacker accuracy at the shadow-optimal threshold $\tau^*$ chosen to maximize balanced shadow accuracy (values near $50\%$ indicate chance).

| Method | MNIST | | | | | | Fashion-MNIST | | | | | |
|---|---|---|---|---|---|---|---|---|---|---|---|---|
| | $AUROC_F$ | $AUROC_R$ | $F-TPR5$ | $R-TPR5$ | $MIA_I$ | $MIA_{II}$ | $AUROC_F$ | $AUROC_R$ | $F-TPR5$ | $R-TPR5$ | $MIA_I$ | $MIA_{II}$ |
| RT | 49.94±0.09 | 50.01±0.03 | 4.85±0.27 | 4.60±0.08 | 100.00±0.00 | 51.81±0.51 | 49.98±0.04 | 49.21±0.27 | 4.89±0.13 | 2.57±0.73 | 100.00±0.00 | 58.44±1.98 |
| PRU | 50.05±0.08 | 50.01±0.03 | 5.09±0.25 | 4.66±0.13 | 100.00±0.00 | 50.91±0.22 | 49.56±0.23 | 49.22±0.24 | 3.49±0.67 | 2.80±0.71 | 100.00±0.00 | 56.99±1.56 |
| BS | 50.01±0.09 | 49.83±0.13 | 5.03±0.26 | 4.19±0.71 | 85.98±27.00 | 50.68±0.24 | 49.61±0.28 | 48.95±0.34 | 3.92±0.49 | 2.15±0.81 | 99.30±1.97 | 55.27±1.29 |
| BE | 49.88±0.12 | 49.97±0.06 | 4.56±0.28 | 4.53±0.43 | 95.05±17.07 | 50.77±0.24 | 48.90±0.69 | 49.21±0.30 | 2.98±0.81 | 1.93±0.64 | 99.31±0.48 | 55.73±0.73 |
| Unroll | 49.99±0.09 | 50.01±0.12 | 5.02±0.31 | 4.76±0.67 | 45.84±25.64 | 50.31±0.24 | 49.90±0.14 | 49.68±0.22 | 4.64±0.90 | 4.12±0.85 | 81.33±28.03 | 50.89±0.92 |
| Unroll-F | 50.05±0.20 | 49.87±0.06 | 5.04±0.22 | 4.41±0.47 | 92.46±12.61 | 50.69±0.35 | 49.61±0.33 | 48.98±0.31 | 4.27±0.96 | 1.07±0.77 | 99.05±1.74 | 53.94±1.06 |
| UNSC | 50.30±0.10 | 49.96±0.03 | 5.00±0.17 | 1.52±2.20 | 100.00±0.00 | 51.24±0.17 | 49.88±0.23 | 49.18±0.21 | 3.70±0.61 | 0.27±0.97 | 100.00±0.00 | 57.05±2.09 |
| SALUN | 50.01±0.07 | 50.01±0.05 | 4.90±0.32 | 4.78±0.11 | 100.00±0.00 | 50.94±0.26 | 49.62±0.14 | 49.68±0.28 | 3.33±0.47 | 2.19±0.55 | 100.00±0.00 | 56.03±1.44 |
| GA | 49.90±0.39 | 49.92±0.07 | 4.95±0.27 | 2.00±1.84 | 84.41±13.51 | 50.48±0.27 | 49.60±0.42 | 49.08±0.27 | 4.50±0.44 | 1.40±1.30 | 98.06±3.53 | 53.47±0.87 |
| Fisher | 49.98±0.08 | 49.97±0.03 | 4.96±0.26 | 4.65±0.12 | 96.48±10.38 | 50.74±0.22 | 49.43±0.26 | 49.13±0.31 | 3.67±0.54 | 2.48±0.79 | 100.00±0.00 | 55.67±1.17 |
| Bad-T | 50.01±0.11 | 50.00±0.04 | 5.12±0.36 | 4.79±0.13 | 28.04±36.22 | 50.33±0.34 | 49.82±0.08 | 49.66±0.16 | 4.30±0.35 | 3.62±0.42 | 95.94±16.00 | 51.85±0.33 |
| SP | 49.93±0.09 | 49.99±0.02 | 4.76±0.24 | 4.57±0.10 | 99.91±0.48 | 50.86±0.14 | 49.93±0.04 | 49.11±0.28 | 4.80±0.18 | 2.48±0.74 | 100.00±0.00 | 55.65±1.28 |
| SCRUB | 49.94±0.11 | 50.02±0.03 | 4.91±0.35 | 4.77±0.14 | 64.47±21.09 | 50.46±0.43 | 49.84±0.07 | 49.43±0.22 | 4.54±0.20 | 3.37±0.66 | 99.48±1.42 | 52.16±0.83 |

| Method | CIFAR-10 | | | | | | CIFAR-100 | | | | | |
|---|---|---|---|---|---|---|---|---|---|---|---|---|
| | $AUROC_F$ | $AUROC_R$ | $F-TPR5$ | $R-TPR5$ | $MIA_I$ | $MIA_{II}$ | $AUROC_F$ | $AUROC_R$ | $F-TPR5$ | $R-TPR5$ | $MIA_I$ | $MIA_{II}$ |
| RT | 50.03±0.07 | 49.21±0.12 | 5.14±0.27 | 0.20±0.76 | 100.00±0.00 | 57.45±0.84 | 50.05±0.06 | 47.30±0.10 | 5.21±0.29 | 0.00±0.00 | 100.00±0.00 | 75.84±0.49 |
| PRU | 49.70±0.11 | 49.38±0.12 | 3.89±0.38 | 2.83±0.45 | 100.00±0.00 | 57.00±0.71 | 49.92±0.08 | 48.21±0.13 | 4.56±0.33 | 0.02±0.01 | 100.00±0.00 | 72.45±0.48 |
| BS | 49.68±0.09 | 49.02±0.13 | 4.14±0.22 | 1.44±0.64 | 99.94±0.13 | 54.42±0.66 | 48.62±0.30 | 45.82±0.15 | 3.61±0.30 | 0.92±0.21 | 97.93±0.94 | 67.17±0.99 |
| BE | 46.92±0.84 | 49.09±0.15 | 2.54±0.60 | 0.03±0.02 | 92.59±2.76 | 56.71±0.66 | 41.65±0.74 | 46.93±0.10 | 0.34±0.12 | 0.01±0.00 | 72.43±2.86 | 74.31±0.45 |
| Unroll | 46.94±0.95 | 49.13±0.17 | 2.83±0.70 | 0.49±0.84 | 82.97±5.75 | 56.43±0.71 | 44.40±0.91 | 47.19±0.12 | 0.41±0.36 | 0.00±0.00 | 36.23±9.84 | 75.02±0.63 |
| Unroll-F | 48.72±0.83 | 48.97±0.18 | 3.77±0.55 | 0.97±0.67 | 92.65±3.35 | 55.26±1.07 | 43.10±0.70 | 46.84±0.19 | 0.75±0.50 | 0.02±0.02 | 57.01±10.27 | 73.61±0.86 |
| UNSC | 49.73±0.13 | 48.99±0.16 | 4.11±0.38 | 0.00±0.00 | 100.00±0.00 | 56.01±0.60 | 49.60±0.14 | 46.66±0.10 | 4.02±0.42 | 0.00±0.00 | 100.00±0.00 | 73.69±0.40 |
| SALUN | 49.80±0.08 | 49.44±0.09 | 4.41±0.29 | 2.23±0.92 | 99.98±0.04 | 53.36±0.77 | 49.96±0.08 | 49.89±0.30 | 4.78±0.36 | 4.78±0.31 | 78.09±11.66 | 50.88±1.55 |
| GA | 49.87±0.08 | 48.97±0.15 | 4.82±0.22 | 1.81±0.70 | 99.69±0.37 | 53.66±0.71 | 45.36±0.63 | 47.08±0.08 | 0.02±0.01 | 0.00±0.00 | 18.21±3.36 | 74.90±0.43 |
| Fisher | 49.30±0.22 | 49.26±0.12 | 3.27±0.38 | 2.50±0.36 | 100.00±0.00 | 56.92±0.66 | 49.45±0.09 | 47.24±0.10 | 3.30±0.39 | 0.01±0.00 | 100.00±0.00 | 74.28±0.41 |
| Bad-T | 49.40±0.16 | 49.24±0.23 | 3.05±0.49 | 1.69±0.97 | 100.00±0.00 | 55.45±0.53 | 49.63±0.08 | 47.92±0.08 | 4.08±0.28 | 3.00±0.13 | 97.73±5.31 | 56.99±0.34 |
| SP | 49.88±0.06 | 49.21±0.13 | 4.63±0.22 | 2.28±0.61 | 100.00±0.00 | 56.24±0.72 | 49.79±0.06 | 47.79±0.08 | 4.33±0.23 | 0.21±0.02 | 99.98±0.04 | 66.56±0.37 |
| SCRUB | 49.87±0.07 | 49.34±0.12 | 4.61±0.28 | 3.01±0.37 | 99.59±2.04 | 53.09±0.52 | 49.66±0.11 | 47.50±0.10 | 4.19±0.27 | 0.89±0.06 | 95.30±2.54 | 62.43±0.44 |

# F  THE USE OF LLMs

We used large language models (LLMs) strictly as writing aids for *language refinement*. Concretely, LLM prompts were limited to grammar correction, concise rephrasing, and minor reorganization of sentences or paragraphs to improve clarity and brevity.

**Scope and limitations.** LLMs were *not* used for ideation, method or theorem development/proofs, algorithm design, experimental setup or tuning, data collection/labeling, result selection, code generation, figure creation, or statistical analysis. All technical content (definitions, theorems, proofs, algorithms, experiments, and conclusions) is authored and validated by the authors.

