# OpenReview forum: "The Illusion of Forgetting: Post-hoc Utility Recovery from Unlearned Models"
_ICLR.cc/2026/Conference — ICLR 2026 Conference Withdrawn Submission_

### Official Review · Reviewer_7rZD · 2025-10-28

**Soundness:** 2
**Presentation:** 2
**Contribution:** 2
**Rating:** 2
**Confidence:** 4

**Summary:**

The paper focuses on class unlearning and, in particular, on researching the possibility of recovering unlearned classes after class unlearning methods. It discusses and evaluates 12  MU methods across 4 datasets and claims to expose a ‘blind spot’ in these works and their evaluations, showing that unlearned classes can be recovered using a proposed technique (which is a black-box method, relying only on model outputs).

This is a well-researched subject and the paper appears to be uninformed by previous work. As such, the paper fails to make clear what, if any, advances are being made with this work. Specifically, prior work has already exposed this claimed new 'blind spot'.

So, given that the issue is not a 'blind spot', the paper should focus on why the proposed method is better than previous ones. And it fails to explore the relevant design space, position itself within it, and quantify/qualify the advantages of the proposed method.

**Strengths:**

1. The approach, showing the recoverability of previously-unlearned classes via an MU algorithm using only a black-box, output-only  method is interesting.

2. The study of 12 algorithms across 4 datasets is comprehensive enough and the results do agree with the claimed ones.

3. The paper is fairly well-written.

**Weaknesses:**

Overall:

As it stands, the paper’s novelty and the significance of its results are questionable.
This is a well-researched subject, and the paper fails to make connections with the fairly large body of work that has addressed the same problem.

Specifically:
1. Key related work is neither cited nor compared against.
In fact, the paper needs a proper/revamped Related Work Section.

Examples of prominent prior research for this problem follow:

-- the work by Bertran et al (NeurIPS24).

-- the work by Ha et al, “Unlearning’s Blind Spots: Over-Unlearning and Prototypical Relearning Attack”, …

More relevant works exist - check the references within the above for related efforts.
These should/could be used as a baseline or discussed at length at the very least, clearly showcasing advances beyond the SOTA work in this area, if any (for exposing the blind spot). Why is the proposed method better? How much better is it?

It **is understood** that this work uses statistical analysis of black-box outputs alone and is different to adversarial attacks research in that regard. But both lines of work expose the same 'blind spot'. So the community is already aware of this problem. Given this, the work **must** quantify and qualify why being based on statistical analysis of black-box outputs alone is preferable, advantageous, etc. In other words, **the emphasis should not be on 'exposing a blind spot'** (as it is now) but on **better methods to do so**. And the work for this line of investigation is missing.

I suspect the answer to the above revolves around being a black-box, output-only method. First, **even for black-box output-only attempts, this is not the first work**, to my knowledge. The IJCAI 2025 paper by Sui et al, achieves the same, no?. But it is neither cited nor compared against.

Even the work "Unlearning Inversion Attacks" by Hu (which is cited) achieves the same (only it requires access to the original model), no? So then readers would benefit a lot if the paper analyzed the 'design space' of black-box vs wiite-box, with access to original model or not, etc. Black-box, output-only attacks are easier'but do not help against more powerful, stronger adversaries with access to weights, and/or to the original model,  etc. All of this should be discussed and not hidden as it is now with specific references to model deployments and different scenarios.

2. The theoretical results are interesting. But they are based on **key assumptions which should be stated more prominently** in the paper. And which **should be explained and justified**. In particular, with respect to their practicality! Given these theoretical results, is it fair to claim that the method can recover unlearned classes? There appears to be a disconnect between claims and what the theory shows.

3. I would urge the authors to consider other benchmarking-like efforts, such as the work by Cadet et al (/https://arxiv.org/pdf/2410.01276) and by Triantafillou et al (arXiv:2406.09073) to compare against MU methods typically used in MU evaluations.

4.  There exist more recent MU methods with better performance than those presented in the set of 12 MU methods evaluated. In fact, the MU methods examined stop at 2024 and do not include well-known MU papers, such as the one by K. Zhao et al (NeurIPS24) and others published in 2024 and in 2025. I urge the authors to do a complete bibliographic search.

5. The experimental results are fairly limited in the dimension of **scale**. Would the results hold on larger **datasets** (ie ImageNet)? Would they hold on larger and different **models** (e.g. vision transformers, such as ViT)?

6. The paper’s title should make crisper the focus of the paper (e.g, class unlearning, in image classification…). This is important as there has been a large number of research papers dealing with the same problem in LLMs, or diffusion T2I models for example.

Overall, the paper has merit and, if positioned and motivated correctly and compared against the proper baselines, with new MU methods, it could be valuable.

**Questions:**

Please see the weaknesses section above.

I remain open to improving my score, assuming the above are addressed.

---

### Official Review · Reviewer_j3BQ · 2025-10-30

**Soundness:** 2
**Presentation:** 2
**Contribution:** 2
**Rating:** 4
**Confidence:** 4

**Summary:**

The paper argues that even after standard “successful” unlearning (near-zero forgotten-class accuracy, strong MIA), the outputs of unlearned models still contain recoverable signals about forgotten classes. It formalizes this with information-theoretic and geometric analyses, and demonstrates a black-box recovery pipeline that restores large amounts of forgotten-class accuracy across 12 unlearning methods and 4 datasets, with minimal hit to retained classes.

**Strengths:**

- The paper proposed a clear, reproducible black-box recovery pipeline that exposes residual forgotten-class signal across many unlearning methods and datasets.
- This provides useful conceptual takeaway for practitioners/evaluators: standard "near-zero FA + good RA" can still leak recoverable utility, motivating stronger evaluations.

**Weaknesses:**

- The scope of the paper is limited to class unlearning on vision classifiers. It would be more interesting to see such analysis on sample-/feature-level unlearning, larger scales, or non-vision tasks.

- Consider a straightforward mitigation: masking/zeroing logits for the forget class or removing those output heads, and this simple mitigation approach can blunt the proposed recovery, but this is obviously not considered as an "ideal" unlearning approach as it doesn’t actually remove any knowledge. If such simple defenses evade your detector, it may suggest the auditing approach is incomplete for practical deployments.

- The theoretical analysis includes heuristic steps (e.g., near-uniform per-class MI share, some conditioning inconsistencies), so the results read as intuition-building rather than tight guarantees.

**Questions:**

Could you assess robustness under common output post-processing (e.g., temperature scaling/calibration, label smoothing, ensembling, etc)? It would help to know when recovery still works and when your detector can flag cases where outputs have been deliberately sanitized.

---

### Official Review · Reviewer_huCs · 2025-10-31

**Soundness:** 3
**Presentation:** 3
**Contribution:** 3
**Rating:** 6
**Confidence:** 4

**Summary:**

The paper discusses that most unlearning algorithm has the “illusion” of unlearning, while the models’ outputs still contain residual information about the forget-set class, which can be recovered by simple methods. The paper shows theoretical insights about first) the fact that there is an unavoidable tension between forgetting and the model’s utility on the retained classes, hence, making complete forgetting almost impossible, second) that unlearning algorithms leave statistical signatures in the models’ output space, and show bounds on residual information in these signatures. The paper also introduces a recovery attack that involves several steps, including identifying the forgotten class using the average probability and variance on the test dataset, and transforming probability values. Experiments on 12 unlearning methods for class unlearning tasks show that forgotten-class predictions can be recovered significantly.

**Strengths:**

**S1**.  The paper studies an important problem. Understanding where and how unlearning methods fail is crucial from many aspects.

**S2**. The paper proposes a novel algorithm that is very intuitive and has been shown to work well for class unlearning.

**S3**. The experiments include examining 12 unlearning algorithms.

**Weaknesses:**

**W1**. Given the scope of the paper, it would be more suitable to emphasize “class unlearning” in the title.  In general, while insightful, the paper has limited applicability as it only addresses class unlearning.

**W2**. The paper does not discuss defense mechanisms. The recovery method uses output probabilities, which are sometimes inaccessible or quantized in real deployments (e.g., APIs returning top-1 labels). The paper does not discuss to what extent the phenomenon persists under restricted output access (e.g., rounded logits, differential privacy noise).

**W3**. Although the paper critiques current evaluation paradigms, it still primarily reports accuracy itself. Further analyses, such as calibration error, mutual-information estimates, or representation similarity, can add some angles to the results.

**W4**. The failure modes you identify are very interesting, but can be presented better. For example, pointing to the results/experiments where these can be spotted.

**W5**. The two-component design (Yeo–Johnson + adaptive thresholding) is intuitive, but there is little analysis isolating their individual contributions. A simple linear scaling or power transform baseline could test whether Yeo–Johnson is truly essential.

**Questions:**

**Q1**. Is the theoretical lower bound measurable in practice? Have you estimated Ires empirically?

**Q2**. Can the authors envision any defenses that destroy these residual traces while maintaining utility?

**Q3**. Can you clarify the practical relevance of Theorem 2? The recovery framework does not seem to use trajectory analysis. Does this theorem have implications for the recovery task that

---

### Official Review · Reviewer_4AUq · 2025-11-01

**Soundness:** 2
**Presentation:** 2
**Contribution:** 3
**Rating:** 4
**Confidence:** 4

**Summary:**

This paper discuss the vulnerability of current machine unlearning methods, whose prediction on the forget data could be recovered by post-processing the predicted logits. This paper covers the class-unlearning case of machine unlearning and the proposed method could recover the residual information from the unlearned model to re-recognize the forget class.

**Strengths:**

1. The paper points out an important issue in machine unlearning, simply relying on the forget accuracy is not enough.

2. The post-process method successfully recovers the accuracy on the forget class, which reveals that there is a way to probe the residual knowledge in the unlearned model.

**Weaknesses:**

1. It seems that the experiments only focus class unlearning, but no experiments on sample unlearning, which means that only a part of samples within a class is forget. As in general, the unlearning could be request at individual level.

2. About theorem 1, the condition is alpha_r > alpha_0; however, in Table 1, most of unlearning methods can not maintain better accuracy on remaining classes than the original model. Therefore, the condition is rarely hold. That is, what is the bound for residual information in most of cases?

**Questions:**

1. It is unclear to me why the authors show OOD results and what does Ours mean there? Is it because the proposed method could be considered as to detect the OOD classes? More elaboration is needed for better clarity.

2. In figure 3b, it shows that even with small set of test set, the proposed method works; however, I am more interesting to know how many classes of samples the proposed method need? As in practices, the user might not know how many classes are trained by the service provider (model), how do the users provide enough samples to justify whether or not the service provider has residual knowledge of the user self?

3. how does kappa in eq 8 affect the identification accuracy?
4. How will those regularization methods, like mixup, cutmix, label smoothing, affect the recovery? As the original probability distribution will be more smooth.

5. Since theorem 1 and 3 are unrelated to the unlearned performance over forget set, is there any way to assess the quality of unlearn method purely based on the performance after unlearning?

6. At line 114, isn't the standard is to be closer to the retrained model?

---

### Note · Authors · 2025-12-01

**Comment:**

Dear Area Chairs and Reviewers,

We would like to express our sincere gratitude to all the reviewers for their thoughtful and constructive feedback on our submission. Your insights have been instrumental in helping us better understand the strengths and weaknesses of our current work, and they have highlighted several important directions for future improvement.

After careful consideration, we have decided to withdraw this submission from ICLR 2026. We greatly appreciate the opportunity to engage with the community on this important topic.

Best regards,

Authors

**Withdrawal Confirmation:**

I have read and agree with the venue's withdrawal policy on behalf of myself and my co-authors.